# Specific sensory neurons and insulin-like peptides modulate food type-dependent oogenesis and fertilization in *Caenorhabditis elegans*

**Shashwat Mishra[1], Mohamed Dabaja[1], Asra Akhlaq[1], Bianca Pereira[1], Kelsey Marbach[1], Mediha Rovcanin[1], Rashmi Chandra[1], Antonio Caballero[2], Diana Fernandes de Abreu[2], QueeLim Ch'ng[2], Joy Alcedo[1]\***

[1]Department of Biological Sciences, Wayne State University, Detroit, United States; [2]Centre for Developmental Neurobiology, King's College London, London, United Kingdom

**Abstract** An animal's responses to environmental cues are critical for its reproductive program. Thus, a mechanism that allows the animal to sense and adjust to its environment should make for a more efficient reproductive physiology. Here, we demonstrate that in *Caenorhabditis elegans* specific sensory neurons influence onset of oogenesis through insulin signaling in response to food-derived cues. The chemosensory neurons ASJ modulate oogenesis onset through the insulin-like peptide (ILP) INS-6. In contrast, other sensory neurons, the olfactory neurons AWA, regulate food type-dependent differences in *C. elegans* fertilization rates, but not onset of oogenesis. AWA modulates fertilization rates at least partly in parallel to insulin receptor signaling, since the insulin receptor DAF-2 regulates fertilization independently of food type, which requires ILPs other than INS-6. Together our findings suggest that optimal reproduction requires the integration of diverse food-derived inputs through multiple neuronal signals acting on the *C. elegans* germline.

**\*For correspondence:**
joy.alcedo@wayne.edu

**Competing interest:** The authors declare that no competing interests exist.

**Editor's evaluation**

In this work the authors perform a rigorous and detailed analysis of the cellular and molecular basis for food type influences on reproduction in *C. elegans*. Their experiments show convincingly that the effects of food type on oogenesis and fertilization are controlled by distinct cellular and molecular pathways. These important findings illuminate a role for insulin-like peptides in linking food type with reproduction and provide a framework for understanding evolutionary forces that may have shaped insulin-like signaling pathways in invertebrates.

## Introduction

The survival of an individual and its species involves reproductive mechanisms that are subject to food-dependent modulation. In the worm *Caenorhabditis elegans*, sensory neurons influence food-dependent survival (*Maier et al., 2010*) and germline physiology (*Dalfó et al., 2012*; *Sowa et al., 2015*; *Aprison and Ruvinsky, 2017*; *Perez et al., 2021*). This suggests the intriguing possibility that sensory perception optimizes survival by modulating the germline in a given environment, which makes it important to understand how the sensory system affects germ cells.

In the germline of the *C. elegans* hermaphrodite, germ cells first undergo spermatogenesis, before switching to oogenesis (*Figure 1A*; reviewed by *Kimble and Crittenden, 2007*). After oogenesis

**Figure 1.** *C. elegans* exhibit early oogenesis and faster fertilization rate in response to a specific bacterial food source. (**A**) The *C. elegans* reproductive developmental program, which depicts the timing of the spermatogenesis-to-oogenesis switch at L4. (**B**) Somatic development was unchanged between animals fed the two *Escherichia coli* food types (n=286 on OP50; n=265 on CS180; p=0.72). (**C–E**) Worms fed OP50 had more progeny (**C**), slower fertilization rates (**D**), and more sperm (**E**) than worms fed CS180. (**F**) Simplified illustration of vulval morphology at various substages of L4, where the nuclei of specific cells are shown (adapted from *Mok et al., 2015*). Red arrowheads indicate the finger-like structures at the sides of the vulva, which migrate ventrally during late L4. Lethargus is the molting period from L4 to young adulthood. (**G**) Early onset of oogenesis was determined by the initial expression of *lin-41::GFP*. Anterior is to the left and dorsal is to the top of each panel. Scale bar is 10 µm. (**H–I**) Worms on CS180 began oogenesis earlier at mid-L4 than worms on OP50, based on the earliest *lin-41::GFP* (**H**) or *oma-1::GFP* (**I**) expression. Early oogenesis onset at mid-L4 is highlighted by a dotted box in these panels and later panels. (**J**) Adults on CS180 had more arrested oocytes than age-matched adults on OP50. Statistical analyses for (**C, D, H and I**) are in *Figure 1—source data 1*. For (**E**), n of animals on OP50 is 14; n on CS180, 12. For (**J**), n of animals on OP50 is 23; n on CS180, 31. Error bars in (**C–D**) represent standard deviation; (**E, J**) SEM. ** indicates p<0.01 and ***, p<0.001.

The online version of this article includes the following source data for figure 1:

**Source data 1.** Statistical analyses of wild-type progeny, fertilization rates, and oogenesis onset on OP50 versus CS180.

begins, oocytes undergo arrest at the diakinesis stage of prophase I (reviewed by *Greenstein, 2005*). Oocytes are released from this arrest in an assembly-line manner by a hormone secreted from sperm, where the oocyte most proximal to the spermatheca matures first (*McCarter et al., 1999*; *Miller et al., 2001*). Once released from arrest, the oocyte is also immediately ovulated into the sperma-theca, where it is fertilized by a sperm (*McCarter et al., 1999*; *Miller et al., 2001*). Thus, in *C. elegans* hermaphrodites, the rate of oocyte maturation determines the rate of fertilization (*McCarter et al., 1999*).

Some of the *C. elegans* sensory neurons have been shown to modulate germ cell proliferation (*Dalfó et al., 2012*; *Aprison and Ruvinsky, 2017*) and oocyte senescence (*Sowa et al., 2015*) in response to food or pheromones. However, it remains unclear whether the same sensory neurons

regulate germline differentiation and fertilization or what signaling mechanisms underlie these sensory influences.

Insulin signaling could mediate the sensory responses to food (reviewed by *Allen et al., 2015*). Worms have 40 insulin-like peptides (ILPs; *Duret et al., 1998*; *Kawano et al., 2000*; *Pierce et al., 2001*; *Li et al., 2003*), many of which are expressed in sensory neurons and could act alone or combinatorially to regulate several physiological programs (*Cornils et al., 2011*; *Chen et al., 2013*; *Ritter et al., 2013*; *Fernandes de Abreu et al., 2014*). Indeed, two ILPs, *ins-3* and *ins-33*, have already been shown to regulate germ cell proliferation (*Michaelson et al., 2010*); however, the cells that secrete these signals that affect the germline are unknown.

In this study, we find that specific sensory neurons and ILPs regulate the timing of oogenesis and fertilization through food type-dependent and -independent mechanisms. In response to a specific food type, we show that the chemosensory neuron ASJ promotes early oogenesis. This ASJ-dependent effect on oogenesis involves ASJ-specific expression of the ILP *ins-6*. In contrast, an olfactory neuron, AWA, regulates food type-dependent fertilization rates through mechanisms that differ from insulin receptor signaling. The insulin receptor pathway alters fertilization rates independent of food cues and employs other ILPs. Thus, our work demonstrates that oogenesis onset and fertilization are subject to different ILP signals and sensory neuron activities, which ensure optimal reproduction within a changing environment.

## Results

### The bacterial food source modulates onset of oogenesis and fertilization rate in *C. elegans*

Previously, we have shown that different *E. coli* strains that serve as food for *C. elegans* can modulate the animals' reproductive physiology, independent of their somatic development (*Maier et al., 2010*). Wild-type *C. elegans* fed two different *E. coli* strains, OP50 and CS180, developed through four larval stages (L1 to L4; *Figure 1A*) at about the same rate (*Figure 1B*), before they molted into young adults. However, *C. elegans* grown on CS180 produced fewer progeny when compared to animals grown on OP50 (*Figure 1C*, *Figure 1—source data 1*). This CS180-dependent phenotype was not because the animals had restricted food intake, since they showed no delay in developmental rates (*Figure 1B*). In fact, CS180-fed animals exhibited faster fertilization rates during the first 24 hr of adulthood (*Figure 1D*, *Figure 1—source data 1*), which is opposite of what would be expected for food level-restricted animals (*Maier et al., 2010*).

In *C. elegans*, the total number of sperm made before the gametogenesis switch to oogenesis can limit the number of progeny (*Ward and Carrel, 1979*). Consistent with this idea, animals fed CS180 not only had fewer progeny (*Figure 1C*, *Figure 1—source data 1*), but also fewer sperm than animals on OP50 (*Figure 1E*). To test the hypothesis that the lower sperm number in *C. elegans* on CS180 was because of an earlier switch from spermatogenesis to oogenesis, we examined when the oogenesis marker *lin-41::GFP* (*Spike et al., 2014*; *Noble et al., 2016*) was first expressed (*Figure 1F to H*, *Figure 1—source data 1*). We found that a significant fraction of worms on CS180 began to express the oogenesis marker *lin-41::GFP* at the mid-L4 stage, which was before the marker was observed in any worms on OP50 (*Figure 1H*, *Figure 1—source data 1*). Worms fed OP50 started to express *lin-41::GFP* only at late L4 (34%), when most animals on CS180 (69%) had already expressed the GFP marker (*Figure 1H*, *Figure 1—source data 1*). To confirm further the effect of the CS180 diet on oogenesis, we also analyzed the initial expression of a second oogenesis reporter, *oma-1::GFP* (*Detwiler et al., 2001*; *Lin, 2003*), on CS180 or OP50. Similar to what we observed with *lin-41::GFP*, worms fed CS180 started to express *oma-1::GFP* at mid-L4 (20%), again before any OP50-fed worms (*Figure 1I*, *Figure 1—source data 1*). The gonads of CS180-fed animals also had more arrested oocytes than the gonads of age-matched adults on OP50 (*Figure 1J*). Thus, our data are consistent with the hypothesis that the fewer progeny on CS180 can reflect earlier oogenesis on this food source. Together with the increased rate of fertilization in CS180-fed animals (*Figure 1D*, *Figure 1—source data 1*), which would depend on the oocyte maturation rate (*McCarter et al., 1999*), our data suggest that CS180 promotes earlier and faster oocyte development.

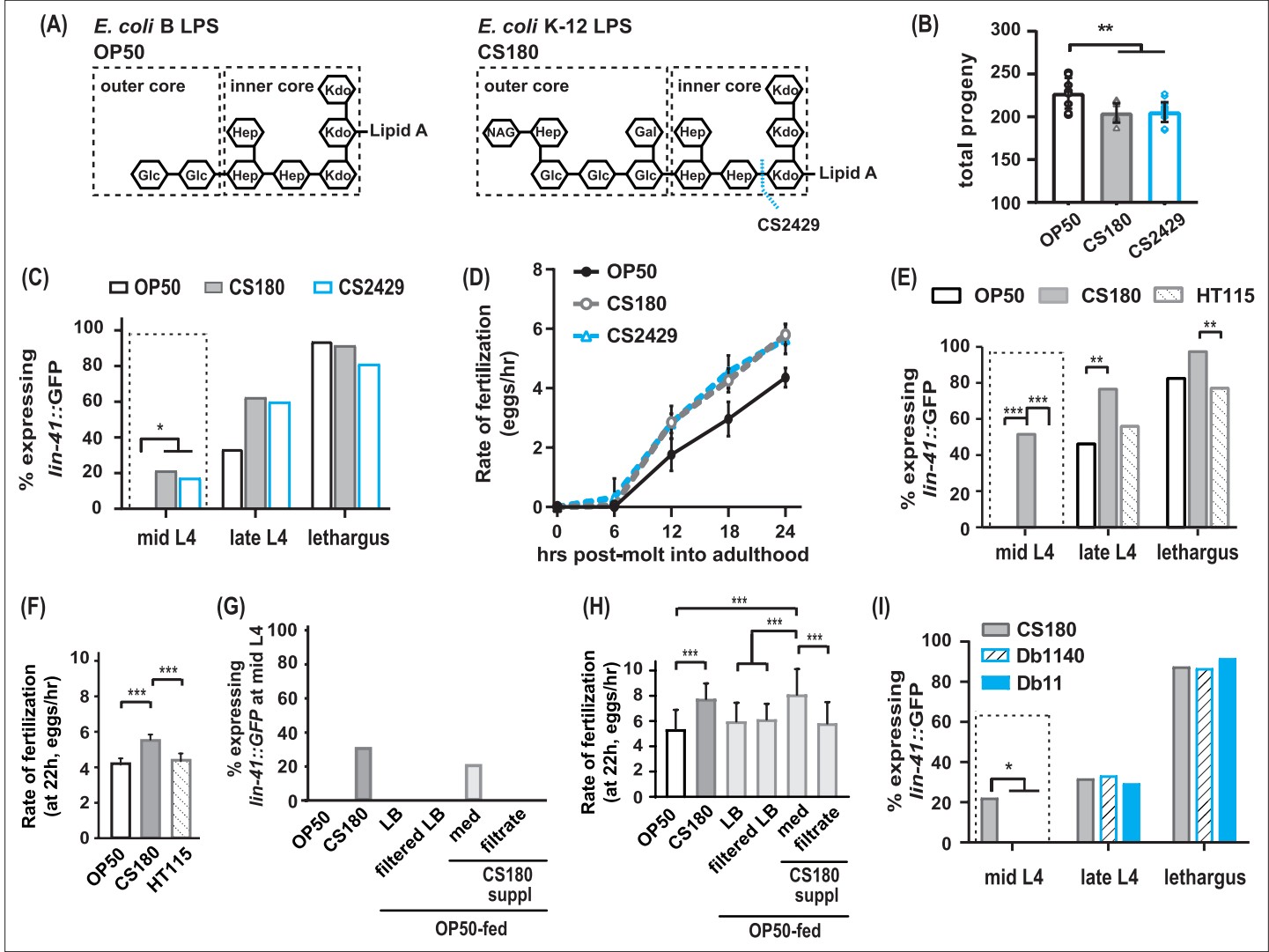

**Figure 2.** The CS180 effects on oogenesis are independent of lipopolysaccharide (LPS) length and absent in HT115 and other bacteria. (**A**) The LPS of *E. coli* B and K-12 strains. CS2429 is derived from CS180 and has a truncated LPS (**Maier et al., 2010**, and references therein). The dotted blue line indicates the site of truncation. (**B–D**) The shorter LPS of CS2429 elicited the same total progeny (**B**), a similar pattern of oogenesis reporter expression (**C**), and the same rate of fertilization (**D**), as the longer LPS of the parent strain CS180. (**E–F**) *C. elegans* oogenesis onset (**E**) and fertilization rates (**F**) on the *E. coli* K-12 HT115 bacteria were compared to those of *C. elegans* on OP50 and CS180. (**G–H**) Oogenesis at mid-L4 (**G**) and fertilization rates at 22 hr (**H**) on OP50 and CS180 were compared to oogenesis onset (**G**) and fertilization rates (**H**) on OP50 that had been supplemented with: (i) CS180-conditioned Luria broth (LB) media (med) or (ii) CS180-conditioned media that was filtered through a 0.45 μm nylon membrane (filtrate). For comparison, worms on OP50 that had been supplemented with LB alone or with LB that was filtered through a nylon membrane are also shown. (**I**) Two strains of the pathogen *S. marcescens*, Db1140 and Db11, did not induce early oogenesis. See **Figure 2—source data 1** for the sample sizes and complete statistical analyses of the data in this figure. Error bars represent standard deviations. * indicates p<0.05; **, p<0.01; ***, p<0.001; and suppl, supplementation.

The online version of this article includes the following source data for figure 2:

**Source data 1.** Statistical analyses of wild-type progeny, oogenesis onset, and fertilization rates on OP50, lipopolysaccharides (LPS) mutant bacteria, HT115, *S. marcescens,* and CS180-conditioned media.

## CS180-derived cues that affect oocyte development are absent from the K-12 HT115 bacteria and *Serratia marcescens*

OP50 is derived from an *E. coli* B strain (**Brenner, 1974**); CS180, from an *E. coli* K-12 strain (**Pradel et al., 1992**). One of the differences between the two bacterial strains is the outer core structures of the lipopolysaccharides (LPS) on their cell walls (**Maier et al., 2010**, and references therein). The OP50 LPS is shorter than the CS180 LPS (**Figure 2A**; **Maier et al., 2010**). To test if this difference in LPS alters oogenesis onset and fertilization rate, we examined wild-type *C. elegans* oocyte development on *E.*

*coli* CS2429, which is a K-12 strain that is derived from CS180 but has a truncated LPS (*Figure 2A*; *Pradel et al., 1992*; *Zhang et al., 2006*). Interestingly, like on CS180, worms on CS2429 still displayed a lower number of progeny (*Figure 2B*, *Figure 2—source data 1*), early oogenesis (*Figure 2C*, *Figure 2—source data 1*), and faster fertilization (*Figure 2D*, *Figure 2—source data 1*). This indicates that this difference in LPS structure is insufficient to affect oocyte development.

Since CS180 is an *E. coli* K-12 strain, we then asked if other K-12 bacteria elicit the same oocyte physiological responses. While worms on the *E. coli* K-12 HT115 bacteria (*Takiff et al., 1989*; *Dasgupta et al., 1998*) produced a similar number of progeny as worms fed CS180 (*Figure 2—source data 1*), HT115-fed worms did not exhibit an early oogenesis onset (*Figure 2E*, *Figure 2—source data 1*) or faster fertilization rates (*Figure 2F*, *Figure 2—source data 1*). This suggests that the cue(s) that cause early oogenesis and faster fertilization (i) might differ from the cue(s) that modulate an animal's number of progeny and (ii) are specific to CS180 and its derivative CS2429.

Thus, we explored the potential nature of the CS180-derived cue(s) that affect oocyte physiology and supplemented OP50-fed worms with CS180-conditioned media. We found that these animals showed early expression of the oogenesis marker at mid-L4 and had faster fertilization rates, which are unlike the unsupplemented OP50-fed controls (*Figure 2G and H*, *Figure 2—source data 1*). However, filtration of the CS180-conditioned media through a nylon membrane, which has a 0.45 µm diameter pore size, removed the cue(s) that promote early oocyte development and fast fertilization (*Figure 2G and H*, *Figure 2—source data 1*). Because the cue(s) were lost by filtration through the nylon membrane filters (*Hasegawa et al., 2003*), the CS180-derived cue(s) that modulate oocyte physiology are unlikely to be small molecules, such as free metabolites.

Because it remains possible that the CS180-derived cue(s) stimulate *C. elegans* innate immunity, which in turn modulate oocyte fate and physiology, we also exposed wild-type worms to two strains of the more pathogenic bacteria *S. marcescens* (*Pradel et al., 2007*). The absence of early oogenesis onset on *S. marcescens* (*Figure 2I*, *Figure 2—source data 1*) suggests that the CS180 responses are not general innate immune responses. Our findings also show that bacterial pathogenicity is insufficient to promote earlier oocyte development.

## Food type modulates oogenesis onset and fertilization rate at different critical periods during development

To identify the critical periods for food type-dependent effects on onset of oogenesis and on fertilization, we grew worms on one food type and switched them to the second food type at different stages of larval development. Worms that were switched from OP50 to CS180 at the first larval stage (L1; *Figure 3—source data 1*) or at the early part of the third larval stage (L3) behaved like worms that were fed CS180 continuously (*Figure 3A and B*, *Figure 3—source data 1*). These worms produced less progeny (*Figure 3A*, *Figure 3—source data 1*) and expressed the oogenesis reporter earlier than animals grown continuously on OP50 (*Figure 3B*, *Figure 3—source data 1*). Since the germ cells of worms transferred to CS180 still responded to this food source, this suggests that the germ cell decision to switch to oogenesis occurs after early L3.

In contrast, animals that were transferred from OP50 to CS180 at late L3 or later developmental stages showed the phenotype of animals that were always grown on OP50 (*Figure 3A and B*, *Figure 3—source data 1*), which suggests that the germ cell decision to switch to oogenesis has already been made by late L3. This is further supported by our findings that the germ cells of worms that underwent the reciprocal switch from CS180 to OP50 responded consistently to OP50 only until early L3, but not at later stages (*Figure 3A and B*, *Figure 3—source data 1*). Thus, food type can only modulate onset of oogenesis before late L3, consistent with a prior study that showed that germ cell fate determination occurs during this stage (*Barton and Kimble, 1990*).

However, the critical window for the food-type effect on fertilization rate is after the early part of the fourth larval stage (L4; *Figure 3C to F*, *Figure 3—source data 1*). Worms switched between one food source to the second food source at early L4 (*Figure 3D*, *Figure 3—source data 1*) had the same fertilization rates as animals consistently fed the second food source. Animals switched at mid-L4 had a delayed and less robust fertilization response to the second food source (*Figure 3E*, *Figure 3—source data 1*), whereas a food switch immediately after the molt into adulthood had no effect on fertilization rates (*Figure 3F*, *Figure 3—source data 1*). Thus, fertilization rate is influenced by food type until just before *C. elegans* molts into a young adult.

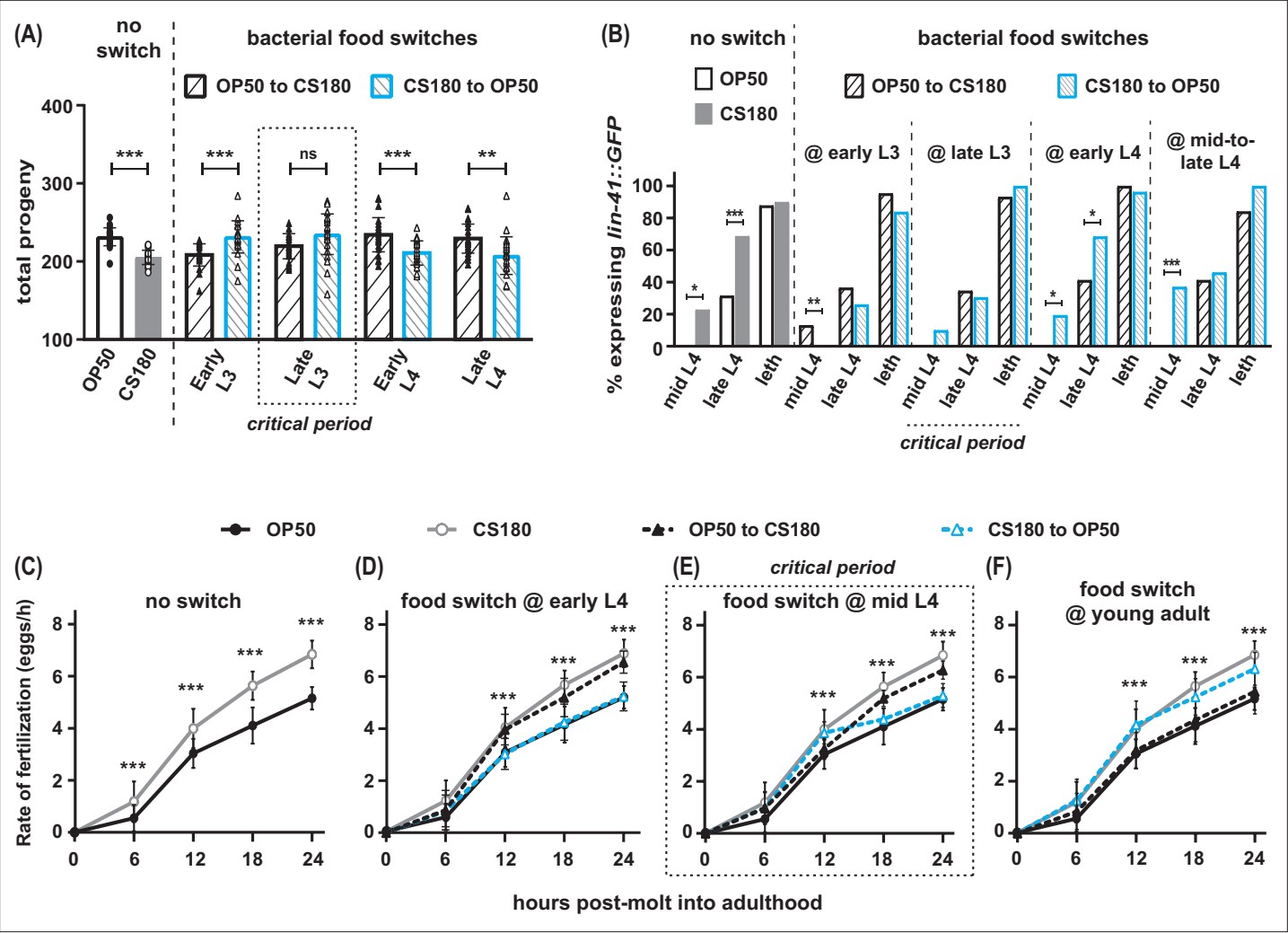

**Figure 3.** Food type modulates oogenesis onset during L3 and fertilization rates after early L4. Wild-type *C. elegans* were shifted between *E. coli* OP50 and CS180 at different stages of larval development. (**A–B**) The critical period for food-type modulation of total progeny (**A**) and oogenesis onset (**B**) is around late L3. (**C–F**) The critical period for modulation of fertilization is after early L4. See *Figure 3—source data 1* for the sample sizes and complete statistical analyses of the data in this figure. Error bars represent standard deviations. * indicates p<0.05; **, p<0.01; ***, p<0.001; ns, not significant; and leth, lethargus.

The online version of this article includes the following source data for figure 3:

**Source data 1.** Statistical analyses of the progeny, oogenesis onset, and fertilization rates in food-switched versus non-switched wild-type animals.

## A specific sensory neuron regulates the food-type differences in total progeny

We have previously shown that sensory perception of food type can alter *C. elegans* longevity (*Maier et al., 2010*), which led us to hypothesize that sensory neurons will also influence the number of progeny the animals produce on the different food sources. Accordingly, we tested mutants that impaired different subsets of sensory neurons. Unlike wild type, the *osm-3* mutant, which has many defective sensory neurons (*Tabish et al., 1995*; *Taylor et al., 2021*), produced the same number of offspring on both OP50 and CS180 (*Figure 4A*, *Figure 4—source data 1*). This suggests that at least some of the *osm-3*-expressing neurons alter the worm's total progeny in response to food type.

Similar to *osm-3* mutants, a mutation in the guanylyl cyclase gene *odr-1*, which impairs a different subset of sensory neurons (*L'Etoile and Bargmann, 2000*; *Taylor et al., 2021*), also produced the same number of progeny on OP50 and CS180 (*Figure 4A*, *Figure 4—source data 1*). However, unlike *osm-3* and *odr-1* mutants, but like wild type, a defect in the transcription factor gene *odr-7*, which

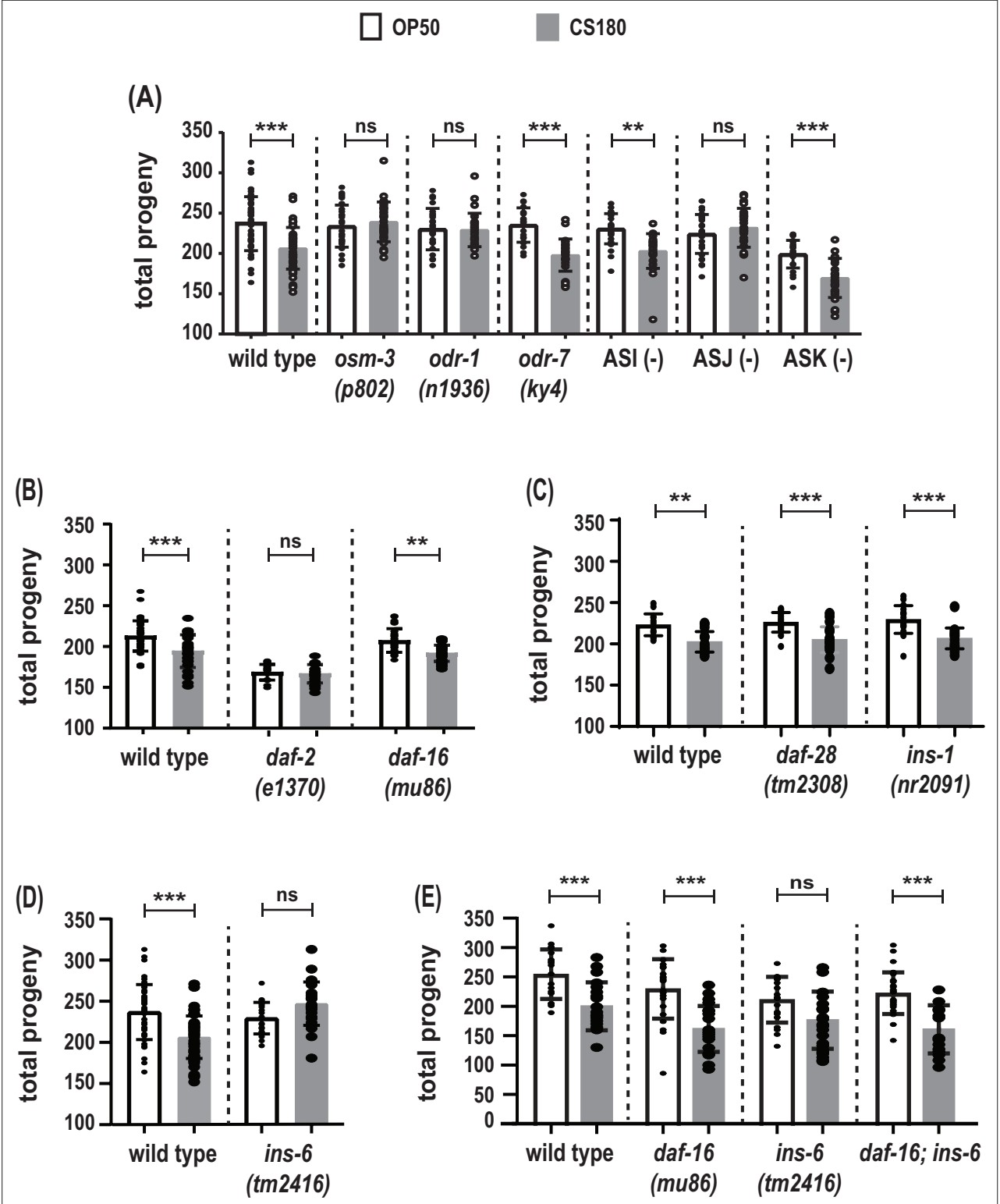

**Figure 4.** The ASJ sensory neuron and insulin-like peptide (ILP) INS-6 alter total progeny in response to food type. (**A**) Food-type differences in total progeny were lost in sensory mutants *osm-3* and *odr-1* and in animals lacking the ASJ neurons. (**B**) Total progeny of the strong reduction-of-function mutant *daf-2(e1370)* and the *daf-16(mu86)* loss-of-function mutant on OP50 and CS180. (**C–D**) OP50- and CS180-dependent progeny of ILP deletion mutants, *daf-28(tm2308)* and *ins-1(nr2091)* (**C**) and *ins-6(tm2416)* (**D**). (**E**) Epistasis analysis between the food type-dependent total progeny phenotypes of *daf-16(mu86)* and *ins-6(tm2416)*. While the animals in (**A**) and (**C–E**) were grown continuously at 25°C, the animals in (**B**) were shifted from 20°C to 25°C at the early L3 stage, because *daf-2* mutants arrest developmentally prior to the L3 stage at 25°C but not at 20°C. See **Figure 4—source data 1** for the

*Figure 4 continued on next page*

*Figure 4 continued*

sample sizes and complete statistical analyses of the data in this figure. Error bars represent standard deviations. ** indicates p<0.01; ***, p<0.001; and ns, not significant.

The online version of this article includes the following source data for figure 4:

**Source data 1.** Statistical analyses of total progeny of different groups of animals on OP50 versus CS180.

causes the specific loss of the olfactory neuron AWA (*Sengupta et al., 1994*), led to more progeny on OP50 than on CS180 (*Figure 4A*, *Figure 4—source data 1*). This suggests that there is some neuronal specificity in the regulation of total progeny.

*osm-3* and *odr-1* mutants share defects in several sensory neurons, including ASI, ASJ, and ASK (*Tabish et al., 1995*; *L'Etoile and Bargmann, 2000*; *Taylor et al., 2021*). Individual ablation of the ASI and ASK neurons did not abolish the food type-dependent differences in total progeny (*Figure 4A*, *Figure 4—source data 1*). Interestingly, loss of the ASJ neurons suppressed the differences in progeny number (*Figure 4A*, *Figure 4—source data 1*) on OP50 versus CS180. Thus, our data show that, unlike ASI, ASK, and AWA, the neuron ASJ modulates the worm's total progeny in response to food cues.

## The ILP *ins-6* also regulates food-type differences in total progeny

The number of *C. elegans* progeny is also subject to regulation by insulin signaling (reviewed by *Murphy and Hu, 2013*), which led us to test whether insulin signaling will modulate the food-type differences in progeny number. Reduction-of-function mutations in *daf-2*, the *C. elegans* insulin-like growth factor receptor (*Kimura et al., 1997*; *Gems et al., 1998*), reduced or abolished the food-type differences in total progeny (*Figure 4B*, *Figure 4—source data 1*). In contrast, a loss-of-function mutation in *daf-16*, the FOXO-like transcription factor that is antagonized by DAF-2 (*Lin et al., 1997*; *Ogg et al., 1997*), still led to food-type differences in the number of progeny on OP50 and CS180 (*Figure 4B and E*, *Figure 4—source data 1*). Together these findings indicate that insulin signaling regulates total progeny in response to food cues.

Because ASJ neurons express at least three potential ILP ligands for the DAF-2 receptor, INS-1 (*Kodama et al., 2006*; *Tomioka et al., 2006*), INS-6 (*Cornils et al., 2011*), and DAF-28 (*Li et al., 2003*; *Taylor et al., 2021*), we tested the effects of these three peptides on progeny number. Deletion of either *ins-1* or *daf-28* did not alter the OP50- or CS180-dependent progeny number (*Figure 4C*, *Figure 4—source data 1*). However, *ins-6* deletion mutants lost the food-type differences in total progeny (*Figure 4D and E*, *Figure 4—source data 1*), which were restored when *daf-16* was also deleted (*Figure 4E*, *Figure 4—source data 1*). Together these data suggest that there are specific ILPs, such as INS-6, that modulate food type-dependent total progeny, which depend on the insulin pathway transcription factor DAF-16.

## ASJ sensory neurons promote early oogenesis on a CS180 bacterial diet

Since sensory neurons modulate total progeny (*Figure 4A*, *Figure 4—source data 1*), we next asked whether the same sensory neurons will also alter the timing of oogenesis in response to food-derived cues. We compared in parallel the onset of oogenesis in *osm-3* mutants and in animals lacking their ASI or ASJ neurons.

Similar to the loss of food-type differences in its total progeny (*Figure 4A*, *Figure 4—source data 1*), *osm-3* mutants failed to exhibit early oogenesis on CS180 during mid-L4 (*Figure 5A*, *Figure 5—source data 1*). This suggests that there are *osm-3*-expressing neurons that promote early oogenesis onset in response to food cues. In contrast, worms lacking ASI neurons showed early oogenesis during mid-L4 (*Figure 5A*, *Figure 5—source data 1*). Yet, the continued low level of oogenesis during late L4 in ASI-ablated worms on CS180 might suggest that ASI plays a minor role in oogenesis and that the food-type differences in total progeny can be uncoupled from the food-type differences in oogenesis onset.

More importantly, ASJ-ablated worms showed no early expression of the oogenesis marker *lin-41::GFP* at mid-L4 and decreased *lin-41::GFP* even during late L4 and lethargus (*Figure 5A*, *Figure 5—source data 1*). ASJ-ablated worms also did not show any mid-L4 expression of a second oogenesis marker, *oma-1::GFP*, which was again different from controls (*Figure 5E*, *Figure 5—source data*

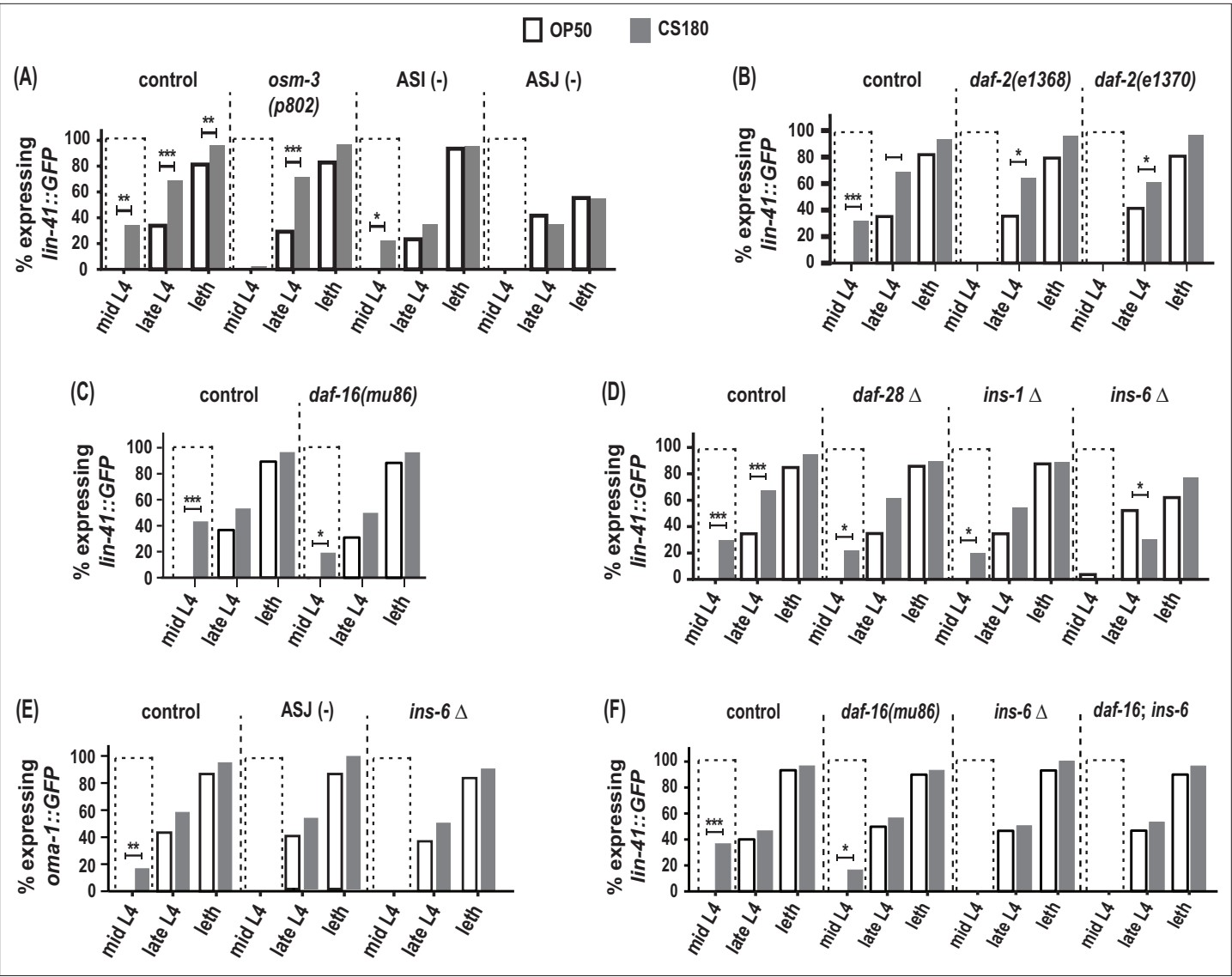

**Figure 5.** ASJ neurons and *ins-6* promote early oogenesis onset. (**A**) Food type-dependent expression of the oogenesis reporter *lin-41::GFP* in control and different sensory mutants. (**B–D**) *lin-41::GFP* expression in *daf-2* reduction-of-function mutants (**B**), *daf-16(mu86)* (**C**), and in insulin-like peptide (ILP) deletion (Δ) mutants (**D**). (**E**) Food type-dependent expression of a second oogenesis reporter, *oma-1::GFP*, in control, ASJ-ablated and *ins-6(tm2416)* worms. (**F**) Loss of *daf-16* did not rescue the delay in oogenesis onset in *ins-6(tm2416)* mutants. Controls and *daf-2* mutants in (**B**) were shifted from 20°C to 25°C as early L3s. In contrast, all other animals were continuously grown at 25°C on OP50 (white bars) or CS180 (gray bars). See *Figure 5—source data 1* for the sample sizes and complete statistical analyses of the data in this figure. * indicates p<0.05; **, p<0.01; ***, p<0.001; and leth, lethargus.

The online version of this article includes the following source data for figure 5:

**Source data 1.** Statistical analyses of oogenesis onset in different groups of animals on OP50 versus CS180.

*1*). Thus, loss of ASJ delayed oogenesis based on the temporal expression of the two oogenesis reporters, *lin-41::GFP* and *oma-1::GFP*. These findings demonstrate that ASJ neurons not only modulate the animal's number of progeny, but also play a major role in promoting early oogenesis.

### *ins-6* promotes food type-dependent early oogenesis independent of the *daf-16*/FOXO transcription factor

How does ASJ promote oogenesis? Again, the ILPs expressed by ASJ are good candidates through which ASJ affects this process, since insulin signaling is an important regulator of oocyte development across metazoans (reviewed by *Das and Arur, 2017*). Indeed, both *daf-2* reduction-of-function mutants lost the early expression of the oogenesis marker *lin-41::GFP* on CS180 (*Figure 5B*, *Figure*

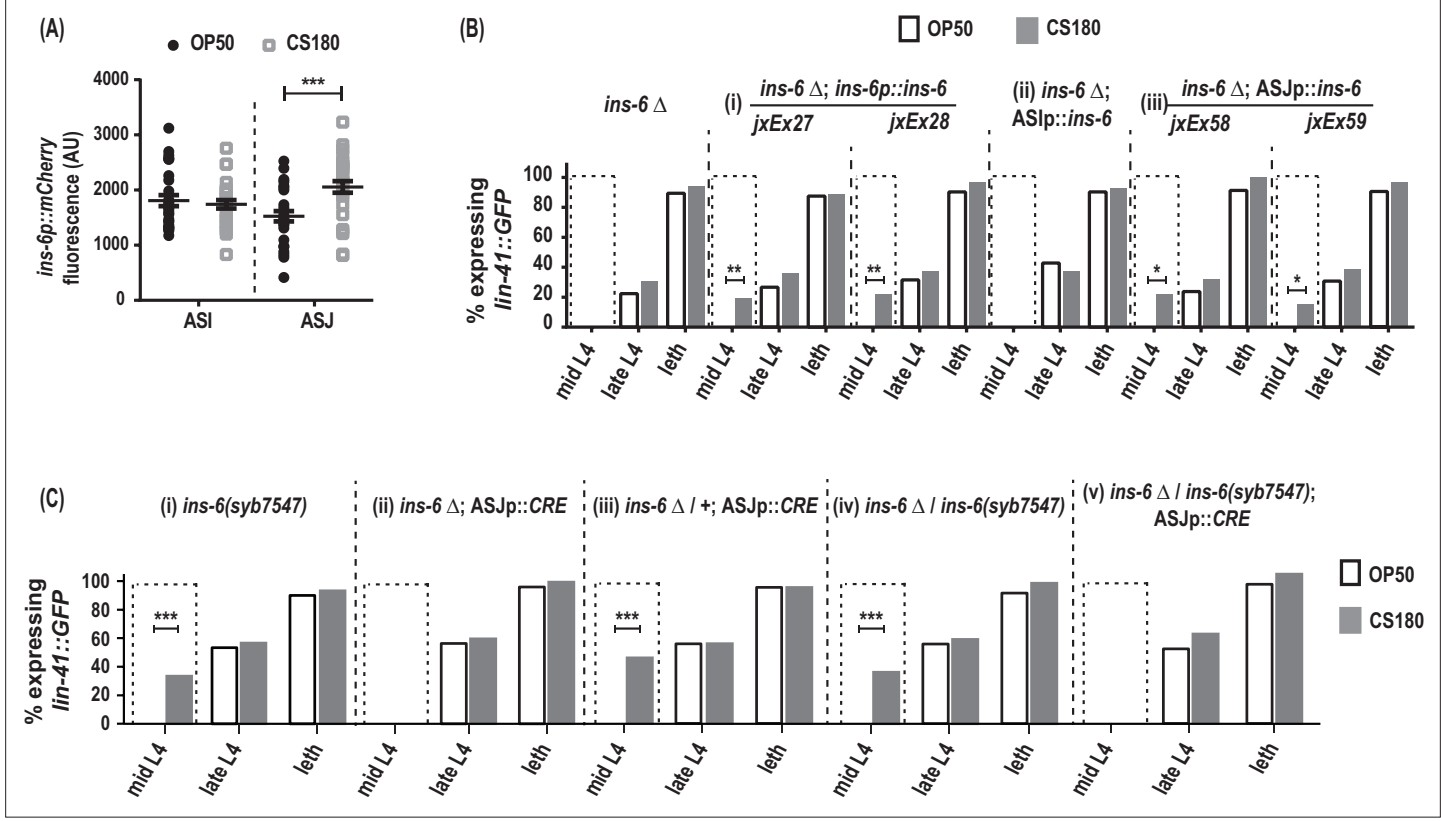

**Figure 6.** *ins-6* acts from ASJ neurons to promote early oogenesis. (**A**) *ins-6p::mCherry* transcriptional reporter *drcSi68* expression in ASI (n=29, OP50; n=29, CS180) and ASJ (n=30, OP50; n=30, CS180) neurons during mid-to-late L3. (**B**) *lin-41::GFP* expression in *ins-6(tm2416)* deletion mutants that were either rescued with: (i) the wild-type *ins-6* genomic locus in two independent lines, *jxEx27* and *jxEx28*; (ii) ASI neuron-specific expression of *ins-6*; or (iii) ASJ neuron-specific expression of *ins-6* in two independent lines, *jxEx58* and *jxEx59*. (**C**) *lin-41::GFP* expression in the (i) *loxP-flanked ins-6(syb7547)* homozygotes, (ii) *ins-6(tm2416)* homozygotes that carry the CRE recombinase transgene that is specifically expressed in ASJ (ASJp::CRE), (iii) *ins-6(tm2416)* heterozygotes that carry the ASJp::*CRE* transgene, (iv) *ins-6(tm2416)/ins-6(syb7547)* transheterozygotes, and (v) animals in which *ins-6* was deleted specifically from ASJ but remains present in ASI. See **Figure 6—source data 1** for the sample sizes and complete statistical analyses of the data in this figure. * indicates p<0.05; **, p<0.01; ***, p<0.001; and leth, lethargus.

The online version of this article includes the following source data for figure 6:

**Source data 1.** Statistical analyses of oogenesis onset in different groups of animals on OP50 versus CS180.

5—source data 1), whereas *daf-16* deletion mutants continued to have early oogenesis on CS180 (**Figure 5C**, **Figure 5—source data 1**). Together these observations suggest that insulin signaling also mediates the food-type effects on worm oogenesis.

To determine whether the same ILP that regulates total progeny again regulates the timing of oogenesis, we assayed the effects of *ins-6* on oogenesis and compared these to the effects of *ins-1* and *daf-28*. Similar to their lack of effect on food type-dependent total progeny (**Figure 4C**, **Figure 4—source data 1**), the deletion of *ins-1* or *daf-28* had no effect on the OP50- or CS180-dependent onset of oogenesis (**Figure 5D**, **Figure 5—source data 1**). However, *ins-6* mutants had no early expression of *lin-41::GFP* (**Figure 5D and F**, **Figure 5—source data 1**) or *oma-1::GFP* (**Figure 5E**, **Figure 5—source data 1**) at mid-L4 on CS180. Thus, our data show that of the three ILPs tested, *ins-6* alone is sufficient to modulate the food type-dependent onset of oogenesis (**Figures 5 and 6**).

Surprisingly, however, the lack of early oogenesis at mid-L4 in *ins-6* mutants cannot be rescued by the loss of *daf-16* (**Figure 5F**, **Figure 5—source data 1**), which suggests that *ins-6* modulates early oogenesis independent of *daf-16*. This is reminiscent of the *daf-16*-independence of *daf-2* in promoting germline meiotic progression in response to food availability (**Lopez et al., 2013**). The *daf-16* independence of *ins-6* in regulating oogenesis also indicates that the mechanisms that underlie food type-dependent oogenesis onset are not completely the same as those that underlie the food-type differences in total progeny.

## ASJ neurons modulate food type-dependent oogenesis onset through INS-6

During the larval stages, *ins-6* is expressed in two sensory neurons, ASI and ASJ (*Cornils et al., 2011*). Using the *drcSi68* transcriptional reporter, which expresses *ins-6p::mCherry* in ASI and ASJ, we asked whether the *ins-6* reporter expression changes when the animals are fed OP50 versus CS180. At mid-to-late L3, the critical period when the food source modulates oogenesis onset (*Figure 3B*, *Figure 3—source data 1*), worms grown on CS180 showed higher *ins-6p::mCherry* expression in ASJ, but not in ASI, unlike worms on OP50 (*Figure 6A*). We also found that ASJ-expressed *ins-6*, but not ASI-expressed *ins-6*, is sufficient to rescue the oogenesis onset phenotype of *ins-6* mutants, comparable to the rescue observed when *ins-6* expression is driven from its own promoter (*Figure 6B*, *Figure 6—source data 1*). These findings suggest that ASJ modulates early onset of oogenesis by producing INS-6.

To confirm that INS-6 does act from ASJ to modulate oogenesis, we next removed the *ins-6* coding sequences from the ASJ neurons, but not from ASI neurons. Loss of *ins-6* from ASJ delayed oogenesis, despite the continued presence of *ins-6* in ASI (*Figure 6C*, *Figure 6—source data 1*). Thus, we show that ASJ-expressed *ins-6* is both sufficient and necessary to promote early oogenesis on the CS180 food source.

## AWA olfactory neurons and insulin receptor signaling regulate food type-dependent fertilization rates through different mechanisms

Next, we asked if sensory neurons also influence fertilization. While the *odr-1* mutation had little effect on food type-dependent fertilization rates (*Figure 7—source data 1*), a mutation in either *osm-3* or another sensory gene, the G protein α subunit *odr-3* (*Roayaie et al., 1998*), suppressed the faster fertilization on CS180 (*Figure 7A and B*, *Figure 7—source data 1*). This again suggests a role for sensory neurons in the food type-dependent modulation of fertilization rates.

Both *osm-3* and *odr-3* are expressed in several chemosensory neurons, including ADF, ASH, ASI, ASK, and ASJ (*Tabish et al., 1995*; *Roayaie et al., 1998*; *Taylor et al., 2021*). Ablation of either ADF or ASH had no effect on the food type-induced variation in fertilization rates (*Figure 7C and D*, *Figure 7—source data 1*). While the global fertilization rates were lower on OP50 and CS180, loss of either ASI or ASK again did not abolish the food-type differences in fertilization (*Figure 7E and F*, *Figure 7—source data 1*). In addition, the ASJ neurons, which modulate onset of oogenesis (*Figure 5A and E*, *Figure 5—source data 1*), did not affect fertilization rates on OP50 versus CS180 (*Figure 7G*, *Figure 7—source data 1*), suggesting that ASJ specifically influences oogenesis, but not fertilization. Thus, our data suggest that other neurons expressing either *osm-3* and/or *odr-3*, either individually or in combination, will modulate food type-dependent fertilization.

We also asked if the *odr-3*-expressing olfactory neurons AWA, AWB, and AWC (*Roayaie et al., 1998*) play a role in this process. While loss of AWB and AWC did not eliminate the fertilization rate differences on OP50 and CS180 (*Figure 7H and I*, *Figure 7—source data 1*), loss of AWA (*Sengupta et al., 1994*; *Yoshida et al., 2012*) suppressed the food-type differences in fertilization (*Figure 7J*, *Figure 7—source data 1*). However, this AWA-dependent regulation of fertilization is independent of the AWA-specific olfactory receptor ODR-10 (*Figure 7—source data 1*; *Sengupta et al., 1996*). This suggests that AWA neurons regulate fertilization in response to food cues that are not ligands of ODR-10.

Next, we asked if AWA and ASJ act synergistically during fertilization. However, we found no synergistic effect between the two neurons in regulating this process (compare *Figure 7G and J to K*, *Figure 7—source data 1*). To determine if AWA neurons specifically regulate fertilization, we then examined when oogenesis begins in animals lacking AWA. We found that absence of AWA had no effect on early onset of oogenesis on CS180 (*Figure 7L*, *Figure 7—source data 1*), similar to its lack of effect on progeny number (*Figure 4A*, *Figure 4—source data 1*). Moreover, we observed no synergistic effect between AWA and ASJ in regulating oogenesis onset (*Figure 7L*, *Figure 7—source data 1*). Thus, on given diets, we show that there are specific neurons that influence distinct aspects of oocyte development and physiology: ASJ for early oogenesis versus AWA for fertilization.

To explore whether AWA regulates fertilization through insulin signaling, we assayed the food type-dependent fertilization rates of insulin signaling mutants. The weak reduction-of-function mutant *daf-2(e1368)* or the null *daf-16(mu86)* mutant had little or no significant effect on fertilization (*Figure 8A*

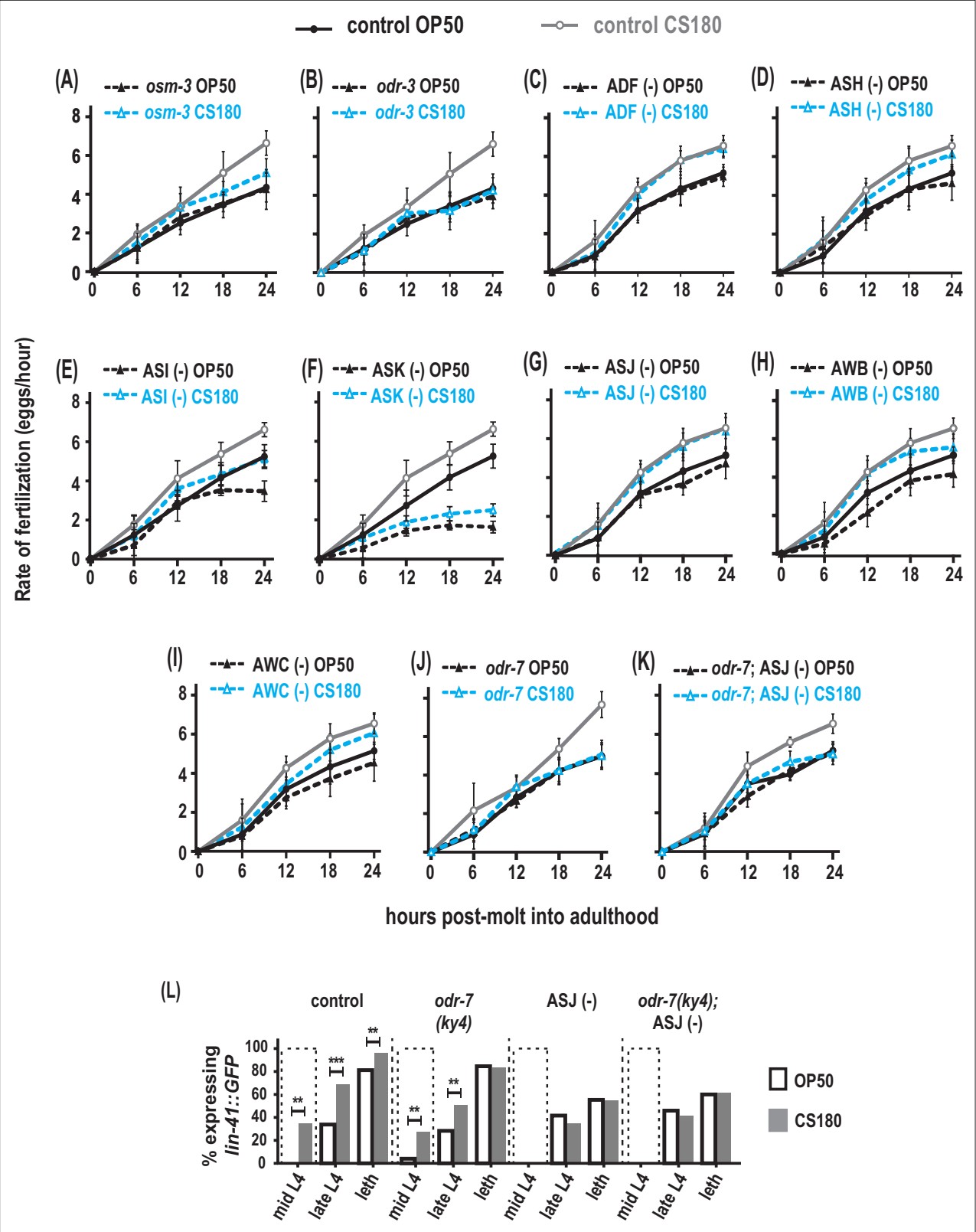

**Figure 7.** AWA olfactory neurons modulate food type-dependent fertilization rates, but not early oogenesis onset. (A–B) Sensory mutations influenced the food-type effects on fertilization rates. (C–I) Animals that lack the sensory neurons ADF (**C**), ASH (**D**), ASI (**E**), ASK (**F**), ASJ (**G**), AWB (**H**), and AWC (**I**). (**J**) Animals lacking the sensory neuron AWA in *odr-7* mutants. (**K**) Animals that lack both AWA and ASJ. Controls for (**G** and **K**) carried the *ofm-1p::gfp* marker in the wild-type background. All other controls (**A–F**, **H–J**) were wild-type worms. (**L**) Food type-dependent expression of the oogenesis reporter

*Figure 7 continued on next page*

*Figure 7 continued*

*lin-41::GFP* in controls or animals lacking AWA and/or ASJ. See **Figure 7—source data 1** for the sample sizes and complete statistical analyses of the data in this figure. ** indicates p<0.01; ***, p<0.001; and leth, lethargus.

The online version of this article includes the following source data for figure 7:

**Source data 1.** Statistical analyses of fertilization rates and oogenesis onset in different sensory-impaired animals on OP50 versus CS180.

and C, *Figure 8—source data 1*), but the stronger reduction-of-function mutant *daf-2(e1370)* (*Lin et al., 1997*; *Gems et al., 1998*) suppressed global fertilization rates (*Figure 8B*, *Figure 8—source data 1*). However, the *daf-2(e1370)* mutant fertilization rate on OP50 still differed from that on CS180 (*Figure 8B*, *Figure 8—source data 1*). This suggests that the DAF-2 receptor modulates overall fertilization rates, but not necessarily in response to food-derived cues. For this reason, the AWA food type-dependent influence on fertilization differs from the effects of insulin receptor signaling on this process.

To test if the ILPs that regulate fertilization also differ from the ILPs that regulate total progeny or oogenesis onset, we measured the effects of *ins-6*, *daf-28*, or *ins-1* on fertilization. A single deletion of any of these ILPs had no effect on fertilization rates (*Figure 8D to F*, *Figure 8—source data 1*), and deletion mutant combinations of these ILPs (*Figure 8—source data 1*) did not exhibit any robust

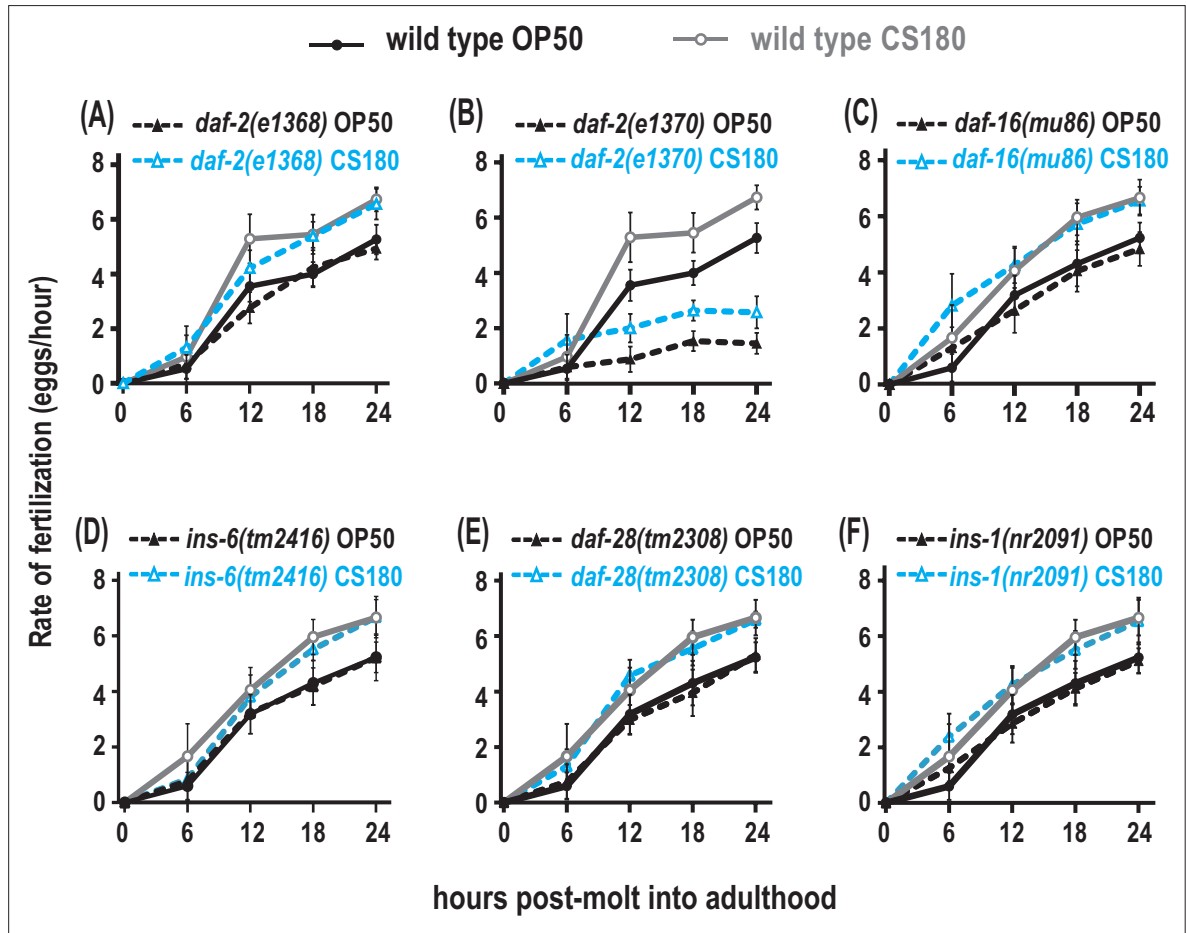

**Figure 8.** Insulin receptor signaling alters global fertilization rates independent of food type. (**A**) The weak reduction-of-function *daf-2(e1368)* mutant. (**B**) The strong reduction-of-function *daf-2(e1370)* mutant. (**C–F**) *daf-16* null mutants and the three insulin-like peptide (ILP) mutants, *ins-6*, *daf-28*, or *ins-1*. Controls and *daf-2* mutants in (**A–B**) were shifted from 20°C to 25°C as early L4s, whereas all other animals were continuously grown at 25°C on OP50 (black closed circles for wild type; black closed triangles for the indicated mutants) or CS180 (gray open circles for wild type; blue open triangles for the indicated mutants). See **Figure 8—source data 1** for the sample sizes and complete statistical analyses of the data in this figure.

The online version of this article includes the following source data for figure 8:

**Source data 1.** Statistical analyses of fertilization rates in insulin signaling-impaired animals on OP50 versus CS180.

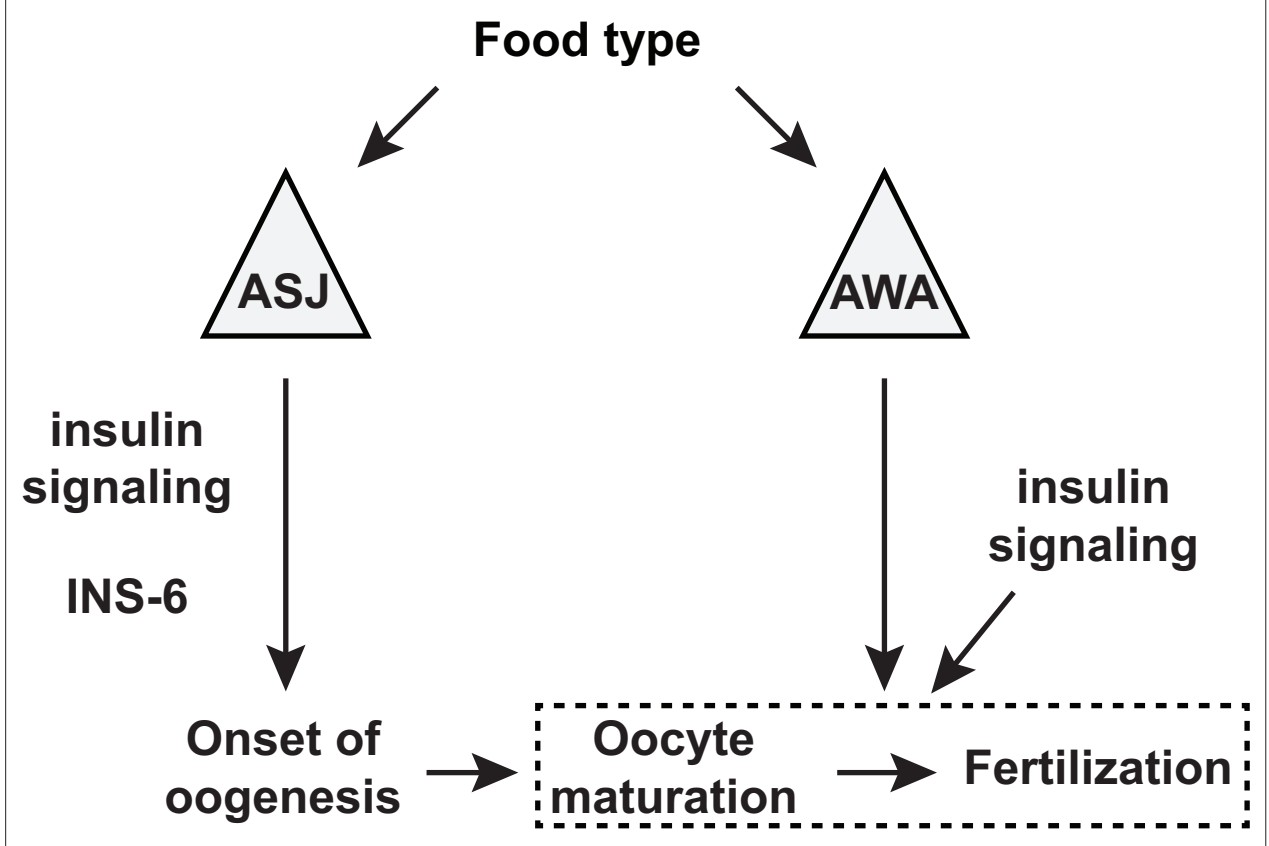

**Figure 9.** A model for the sensory modulation of food type-dependent oogenesis onset and fertilization. Insulin-like peptide (ILP) INS-6 acts from ASJ neurons to promote early onset of oogenesis. Olfactory neuron AWA promotes oocyte fertilization through mechanism(s) distinct from DAF-2 insulin receptor signaling, which in turn regulates fertilization independent of food type through the activities of other ILP(s).

phenotypes that mirrored those of the strong *daf-2(e1370)* mutant phenotypes. This implies that there are multiple ILPs that act through the DAF-2 receptor to regulate fertilization. Notably, our findings indicate that the set of ILP activities that regulates fertilization is distinct from the set of ILPs that regulates onset of oogenesis.

## Discussion

Germline development must be coordinated with existing environmental resources to optimize population survival. The sensory system is an important interface between the environment and an animal's physiology, which suggests a role for the sensory system in germline biology. Not surprisingly, sensory neurons have been implicated in germline proliferation or senescence in *C. elegans* (*Dalfó et al., 2012*; *Sowa et al., 2015*; *Aprison and Ruvinsky, 2017*) and potentially in gamete development in mammals (*Boehm et al., 2005*; *Yoon et al., 2005*). Here, we show that distinct sensory neurons acting through insulin-dependent and -independent signaling mechanisms influence specific steps in oocyte development and physiology in response to food-derived cues. Thus, discrete aspects of oocyte biology—oocyte fate determination and successful oocyte fertilization—are subject to different sets of cues that are sensed by different cells that work together to control the production of an animal's total progeny.

### The sensory neurons that regulate oogenesis onset differ from those that affect fertilization

Unlike the olfactory neuron AWA, the chemosensory neuron ASJ promotes early onset of oogenesis (*Figure 7L*, *Figure 7—source data 1*, *Figure 9*). This ASJ-dependent effect does not appear to be a general response to stress, like pathogen stress (*Figure 2I*, *Figure 2—source data 1*), but a response

to specific food cues that are likely not small metabolites (*Figure 2G*, *Figure 2—source data 1*). In contrast, the olfactory neuron AWA promotes the increased fertilization rates on the CS180 food source, but not ASJ and the other neurons that we have so far tested (*Figure 7C to K*, *Figure 7—source data 1*, *Figure 9*). Together these data suggest that the sensory cue(s) that promote early oogenesis differ from the sensory cue(s) that affect fertilization.

While AWA has a major role in food type-dependent fertilization, it is most likely not the only sensory neuron involved in this process. The *osm-3* mutation, which impairs several sensory neurons (*Tabish et al., 1995*), also partly suppresses the faster fertilization on CS180 (*Figure 7A*, *Figure 7—source data 1*). This raises the likelihood that other neurons regulate fertilization. In fact, we find that worms lacking ASI, ASK, and AWB neurons fertilize at a slower rate at certain or at all time points on OP50 and CS180, when compared to wild type (*Figure 7E, F, and H*, *Figure 7—source data 1*). However, these animals still exhibit significant food-type differences in their fertilization rates (*Figure 7E, F, and H*, *Figure 7—source data 1*). This means that ASI, ASK, and AWB are not required to regulate food-type effects on fertilization, but might have a role that are independent of food. A mechanism that includes several neurons to regulate fertilization would be expected, since it is a dynamic process that proceeds for several days. This makes it necessary for the worms to sense environmental changes continuously and adjust their reproductive physiology accordingly. Thus, the involvement of multiple sensory neurons would make for a more sensitive and efficient physiology.

Interestingly, at least one of the sensory neurons that inhibits germline proliferation (*Aprison and Ruvinsky, 2017*), ASJ, also promotes food type-dependent early oogenesis (this work). This suggests that this sensory neuron might regulate oogenesis onset by regulating the first switch that occurs in the *C. elegans* germline, mitosis (i.e., proliferation) to meiosis (i.e., differentiation) (*Kimble and Crittenden, 2007*). Consistent with this hypothesis, we demonstrate that food type can only modulate onset of oogenesis before the late L3 stage (*Figure 3B*, *Figure 3—source data 1*), which coincides with the time before germ cells enter meiosis (*Kimble and Crittenden, 2007*). Thus, some of the sensory neurons that influence the germline might have more than one function at different stages of germ cell development.

## Oocyte development requires distinct insulin-like signaling mechanisms

Insulin signaling is an important regulator of oogenesis, where impaired signaling leads to reproductive deficits across many species (reviewed by *Das and Arur, 2017*). Here, we find that specific ILPs regulate discrete stages of oocyte physiology. The ILP INS-6 acts from the ASJ neuron to modulate food type-dependent early oogenesis (*Figure 6B and C*, *Figure 6—source data 1*, *Figure 9*), whereas it appears to have little or no effect on oocyte fertilization rates (*Figure 8D*, *Figure 8—source data 1*). Although insulin receptor DAF-2 signaling does not appear to mediate the AWA neuron-dependent food-type effects on fertilization, DAF-2 does regulate overall fertilization independent of food sources (*Figure 8B*, *Figure 8—source data 1*, *Figure 9*), where its ILP ligands remain to be identified. Since fertilization depends on oocyte maturation (*McCarter et al., 1999*), our data also suggest that different ILPs regulate specific stages of oogenesis—onset versus maturation.

Our findings further suggest that at least one of these ILPs, INS-6, can act independent of DAF-16, a known transcriptional effector of the DAF-2 receptor (*Figure 5F*, *Figure 5—source data 1*). Distinct insulin-like signaling mechanisms—*daf-16*-independent and *daf-16*-dependent—might be necessary for optimal oogenesis and fertilization. A worm's number of progeny will likely be impacted by the timing of the spermatogenesis-to-oogenesis switch, as well as by successful fertilization. Hence, distinct ILP signaling mechanisms that can respond to disparate environmental cues might also explain the variability in the number of progeny of *daf-16* mutants (*Figure 4B and E*, *Figure 4—source data 1*).

ILPs might encode and integrate diverse cues that influence oocyte development. Considering the pleiotropic effects of insulin signaling on metazoan reproduction (*Das and Arur, 2017*), it is possible that the multiple ILPs present in other animals (*Allen et al., 2015*) will also have specific roles in coordinating oogenesis with fluctuations in a given environment. Thus, our study suggests that successful fertilization will require the careful integration of multiple environmental inputs through ILPs acting on the germline from neuronal, and potentially non-neuronal, cells.

## Materials and methods
### Worm strains and culture
General maintenance

*C. elegans* were cultured on NGM agar plates (*Brenner, 1974*) seeded with *E. coli* OP50, CS180, CS2429, or HT115 or with *S. marcescens* strains. All bacterial strains were initially cultured in Luria broth (LB) medium to logarithmic phase, prior to seeding the NGM agar plates. All animals were well-fed for at least two generations before any phenotypic analyses were conducted. All experiments were performed at 25°C, except for those that included the *daf-2(e1368)* or *daf-2(e1370)* temperature-sensitive mutations. *daf-2* mutants grown as eggs until the L1 stage at 25°C become developmentally arrested (*Gems et al., 1998*). Thus, these mutants were first grown at 20°C, the permissive temperature, and shifted to 25°C before the critical periods defined for food-dependent modulation of oogenesis onset or fertilization, along with their corresponding controls. See *Supplementary file 1* for the list of genotypes of the strains used in this study, which include the neuronally ablated animals. All strains generated in this study are available upon request.

### Generation of the *lin-41::GFP daf-16(mu86)* recombinant

To recombine *lin-41::GFP* with the *daf-16(mu86)* deletion on chromosome I, the strain carrying *lin-41(tn1541[GFP::tev::s::lin-41])* (*Spike et al., 2014*) was crossed to a *daf-16(mu86)* homozygote. Among the subsequent progeny of the *lin-41::GFP+/+daf-16* cross-progeny, we identified 1 recombination event out of 50 chromosomes, which was homozygous for *lin-41::GFP* and heterozygous for *daf-16(mu86)*. This animal was allowed to reproduce to isolate the homozygous *lin-41::GFP daf-16(mu86)* recombinant strain.

### Generation of the *loxP*-flanked *ins-6* strain, *syb7547*

A region of the *ins-6* genomic locus, which begins at the translation start codon in exon 1 until the translation stop codon in exon 2, was replaced with the *ins-6* open reading frame (ORF) through CRISPR/Cas9 technology by SunyBiotech (Fujian, China). The inserted ORF, which lacks intron 1, was flanked at both ends with the *loxP* sequence, 5'-ata act tcg tat agc ata cat tat acg aag tta t-3'. The first *loxP* sequence was inserted 3 bases upstream of the start codon, whereas the second *loxP* sequence was inserted immediately after the stop codon. Two synonymous nucleotide changes were introduced into the inserted ORF to prevent its cleavage by Cas9 during the integration step: L14L (CTC to CTG) and C96C (TGC to TGT). *syb7547* was outcrossed four times to the *lin-41(tn1541[GFP::tev::s::lin-41])* strain (*Spike et al., 2014*) before analysis in the presence or absence of the *trx-1p::nCRE* recombinase transgene (ASJp::*CRE*). Like animals that have a wild-type *ins-6* genomic locus, *ins-6(syb7547)* animals showed an earlier onset of oogenesis on CS180 during mid-L4 (*Figure 6C*, *Figure 6—source data 1*) in the absence of the *trx-1p::nCRE* recombinase.

### Generation of the ASJp::*CRE* recombinase strain

We drove *nCRE* recombinase (gift of Mario de Bono) from the *trx-1* promoter (*Miranda-Vizuete et al., 2006*; *Cornils et al., 2011*), with an *unc-54* 3' UTR from pPD95.77 (gift of Andrew Fire; pQZ116). The plasmid pQZ116 was introduced into a *lin-41(tn1541[GFP::tev::s::lin-41]); ins-6(tm2416)* animal, as an extrachromosomal array.

## Measurement of the rate of development

L1 larvae that hatched within a 2 hr period on each bacterial food source were collected and allowed to grow at 25°C for 32 hr on the same food source. The number of animals that were L1, L2, L3, or L4 larvae or adults were counted and represented as a percentage of the total population. The chi-square test was used to compare the stage distributions of worms fed the different bacteria.

## Measurement of sperm and arrested oocytes

To count the number of sperm, 2- to 3-hr-old adult OD95 hermaphrodites that carry the transgene *pie-1p::mCherry::his-58* (*McNally et al., 2006*) were anesthetized on a 1% agarose pad that contained 6 mM sodium azide. Series of Z-stack images of the spermathecae were captured at 100 ms as the

exposure time, using a Nikon Eclipse Ti-E microscope and a Photometrics Coolsnap Myo cooled digital camera. The number of sperm was counted as red fluorescent puncta across the entire stacks.

To measure the number of arrested oocytes, young wild-type adult hermaphrodites were anesthetized as described above. The gonads were imaged under differential interference contrast microscopy. Counts from only one gonad arm were reported per animal. The Student's t-test was used to compare gamete counts between animals.

## Measurement of total progeny

Individual young adult hermaphrodites were grown on a 35 mm plate seeded with the indicated bacteria. The mother was transferred to a new plate every 12–24 hr until reproduction ceased entirely. Total progeny that hatched were counted and reported. At least 10 animals were tested in each trial, and each strain studied was assayed in multiple independent trials. To analyze the *daf-2* mutants and their corresponding controls, worms were first grown at 20°C, before they were transferred to 25°C at the early L3 stage. Statistical comparisons across multiple groups were determined by one-way ANOVA, followed by Tukey's correction.

## Imaging the oogenesis reporters

Control and mutants carrying *lin-41::GFP* (*Spike et al., 2014*) or *oma-1::GFP* (*Lin, 2003*) were imaged in parallel at ×400 magnification, using a Nikon Eclipse Ti-E microscope and a Photometrics Coolsnap Myo cooled digital camera or a Nikon Eclipse Ni-U microscope and a Photometrics Coolsnap ES2 cooled digital camera. The exposure time was set at 500 ms, and the presence or absence of GFP was reported at different stages. The stages of the animal were determined by observing the vulval morphology under 400×, as described by *Mok et al., 2015*, and illustrated in *Figure 1F*. Based on the substages described by *Mok et al., 2015*, early L4 corresponded to the L4.0 to L4.2 substages; mid-L4, to the L4.3 to L4.5 substages; late L4, to the L4.6 to L4.8 substages, when the finger-like structures that formed at the sides of the vulva (red arrowheads in *Figure 1F*) had started to migrate ventrally; and lethargus to the L4.9 substage. To analyze the *daf-2* mutants and their corresponding controls, worms were grown at 20°C and transferred to 25°C at the early L3 stage and then imaged at different subsequent stages. Statistical comparisons across multiple groups used the Fisher's exact test.

## Imaging the *ins-6p::mCherry* transcriptional reporter

### Generation of the *ins-6p::mCherry* transcriptional reporter

The mCherry gene was flanked with 1.64 kb sequences upstream of the *ins-6* start codon and 7.57 kb sequences downstream of the *ins-6* stop codon. This reporter was cloned into a MosSCI vector for integration (pQL147) at the ttTi5605 site of chromosome II (*Frøkjaer-Jensen et al., 2008*) to generate the *ins-6p::mCherry drcSi68* worms.

### Imaging of *drcSi68* worms

The worms were grown continuously on OP50 or CS180. Worms at the mid-to-late L3 stage were imaged at ×1000 magnification, using a Nikon Eclipse Ni-U microscope and a Photometrics Coolsnap ES2 cooled digital camera. The exposure time was set at 50 ms. Fluorescence intensities were quantified using a built-in fluorescence quantification algorithm (NIS-Elements, Nikon Instruments, Inc). Statistical analyses used two-way ANOVA and Tukey's correction.

## Measuring the rates of fertilization

For every trial, 10–12 young adult worms were grown singly for a specified number of hours. At the end of each time point, the mother was bleached in a 5% sodium hypochlorite solution to release and count the fertilized embryos within its uterus. We also counted the number of hatched progeny that were laid on the plates. The total number of oocytes fertilized at each time point was the sum of the embryos within the uterus and those laid on the plates. We then divided the total number of fertilized oocytes by the duration of the collection to calculate the rate of fertilization at that specific time point. To analyze the *daf-2* mutants and their corresponding controls, worms were first grown at 20°C and transferred to 25°C at the early L4 stage. Statistical comparisons across multiple groups were performed by two-way ANOVA and Tukey's correction.

To compare the food type-dependent changes in fertilization rates between wild type and different groups of animals, we normalized the 12 to 24 hr fertilization rates on CS180 to the 12 to 24 hr fertilization rates on OP50 of a given strain. The resulting normalized means were then compared to the food type-dependent change in fertilization rates of wild-type controls. Thus, p values were calculated by obtaining the t-score after comparing the percent change in wild type on CS180 versus OP50 to the percent change for the specific mutant on CS180 versus OP50.

## Food-switch experiments

*C. elegans* were grown on each bacteria and then transferred to the second bacteria at different stages. The early L3, late L3, early L4, and late L4 were determined based on the developmental rates of the animals on OP50 and CS180. L1 larvae that hatched within a 1 hr period at 25°C were collected and allowed to develop further at this temperature on the first bacteria for a certain duration before being shifted to the second bacteria: 20 hr for early L3, 24 hr for late L3, and 26 and 30 hr, respectively, for early and late L4. The early and late L4 stages were also confirmed by examining vulval morphology. To minimize contamination from the earlier bacteria, worms were allowed to roam on the second bacteria for 45–60 min and then transferred to a fresh plate seeded with the second bacteria. After this transfer, worms were assayed for total number of progeny, fertilization rate, or the expression of the *lin-41::GFP* oogenesis reporter.

To compare the food type-dependent changes in fertilization rates during the food switches, we normalized the 12 to 24 hr fertilization rates of the CS180-to-OP50 switched animals to the 12 to 24 hr fertilization rates of the OP50-to-CS180 switched animals. The p values of the subsequent normalized means were calculated by obtaining the t-score after comparing the percent change in food-switched animals to the percent change in non-switched controls on CS180.

## Preparation of CS180-conditioned media for supplementation experiments

For each experimental trial, we grew a single colony of *E. coli* CS180 in 10–25 ml LB to an optical density of 0.5–0.6 at 600 nm (logarithmic growth phase) at 37°C. We then centrifuged 3–4 ml of the CS180 culture at 2380 × *g* at 4°C for 15 min to remove the bacterial pellet and isolate the CS180-conditioned medium. We added 100 µl of the CS180-conditioned medium to each lawn of *E. coli* OP50 on 35 mm NGM agar plates to determine the effects of the medium on oogenesis onset and oocyte fertilization rates.

In addition, we passaged the CS180-conditioned medium through a sterile nylon membrane syringe filter with a 0.45 µm diameter pore size. We added 100 µl of the collected filtrate to a different lawn of OP50 on a 35 mm NGM agar plate to test whether the CS180-derived cue(s) that modulate oocyte physiology are small, free metabolites. To verify that the potential loss of CS180 cue(s) was not due to the cue(s) adsorbing to the membrane, we pre-incubated 3–4 ml of the CS180-conditioned medium with a postage stamp-sized 0.45 µm diameter nylon membrane. The pre-incubation was carried out for 10 min at room temperature using a nutator set at low speed. We also added 100 µl of the pre-adsorbed medium to other lawns of OP50 on 35 mm NGM agar plates. Pre-adsorption of the CS180-conditioned media with a nylon membrane filter still led to faster fertilization, similar to CS180-fed controls (*Figure 2—source data 1*).

## Statistical analysis

Statistical analyses were done using GraphPad Prism 8 software. For more details, refer to above and the legends of the figures and their source data files.

## Acknowledgements

We thank S Mitani, the Caenorhabditis Genetics Center (funded by NIH P40 OD010440), J Apfeld, T Chen, M de Bono, D Ferkey, A Fire, M Goodman, J Klena, and P Sengupta for strains used in this study. We also thank S Kuchin of University of Wisconsin-Milwaukee for generously assisting us in the statistical analyses of total fertilization rates of different worm genotypes on different food sources (*Figures 3, 7 and 8 source data files, sheets on normalized fertilization rates*). We further thank A Shevade and M Salsaa for comments on the manuscript. This work was supported by a Wayne State University Graduate Research Assistantship to SM, an ERC Starting Investigator Grant (NeuroAge

242666) and a Research Councils UK Fellowship to QC, and by Wayne State University, the Alcedo family, and NIH (R01 GM108962) to JA.

## Additional information

### Funding

| Funder | Grant reference number | Author |
|---|---|---|
| Wayne State University | Graduate Research Assistantship | Shashwat Mishra |
| European Research Council | NeuroAge 242666 | QueeLim Ch'ng |
| Research Councils UK | | QueeLim Ch'ng |
| National Institute of General Medical Sciences | R01 GM108962 | Joy Alcedo |

The funders had no role in study design, data collection and interpretation, or the decision to submit the work for publication.

### Author contributions

Shashwat Mishra, Conceptualization, Resources, Data curation, Formal analysis, Validation, Investigation, Visualization, Methodology, Writing – original draft, Project administration, Writing – review and editing; Mohamed Dabaja, Kelsey Marbach, Resources, Formal analysis, Validation, Investigation; Asra Akhlaq, Mediha Rovcanin, Rashmi Chandra, Formal analysis, Validation, Investigation; Bianca Pereira, Formal analysis, Validation, Investigation, Resources; Antonio Caballero, Diana Fernandes de Abreu, QueeLim Ch'ng, Resources; Joy Alcedo, Conceptualization, Resources, Data curation, Formal analysis, Supervision, Funding acquisition, Validation, Investigation, Visualization, Methodology, Writing – original draft, Project administration, Writing – review and editing

### Author ORCIDs

QueeLim Ch'ng ⓘ http://orcid.org/0000-0003-1941-3828
Joy Alcedo ⓘ http://orcid.org/0000-0002-5279-6640

### Decision letter and Author response

Decision letter https://doi.org/10.7554/eLife.83224.sa1
Author response https://doi.org/10.7554/eLife.83224.sa2

## Additional files

### Supplementary files
- MDAR checklist
- Supplementary file 1. List of strains used in the study.

### Data availability

All data generated or analyzed during this study are included in the manuscript and supplemental files: *Supplementary file 1* for the strain list used in the study and source data files to Figures 1 to Figure 8.

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
