## [Editor Report]

In this work the authors perform a rigorous and detailed analysis of the cellular and molecular basis for food type influences on reproduction in *C. elegans*. Their experiments show convincingly that the effects of food type on oogenesis and fertilization are controlled by distinct cellular and molecular pathways. These important findings illuminate a role for insulin-like peptides in linking food type with reproduction and provide a framework for understanding evolutionary forces that may have shaped insulin-like signaling pathways in invertebrates.

---

## [Decision Letter]

**Decision letter after peer review:**

Thank you for submitting your article "Specific sensory neurons and insulin-like peptides modulate food type-dependent oogenesis and fertilization in *Caenorhabditis elegans*" for consideration by *eLife*. Your article has been reviewed by 3 peer reviewers, including Patrick J Hu as the Reviewing Editor and Reviewer #1, and the evaluation has been overseen by Piali Sengupta as the Senior Editor.

Essential revisions:

1. Provide more insight into bacterial factors that might be contributing to the CS180 oogenesis phenotype;

2. Perform oogenesis experiments using an additional readout distinct from the lin-41::GFP reporter;

3. Strengthen evidence in support the roles of ASJ and INS-6 in mediating the CS180 early oogenesis phenotype (e.g., ASJ-specific deletion of ins-6);

4. Address the issue of aberrant kinetics of lin-41::GFP expression in the ASJ ablation strain (potentially with a different oogenesis readout).

*Reviewer #1 (Recommendations for the authors):*

1. Lines 84-85: I think the conclusion that the decreased progeny phenotype on CS180 is a readout for oogenesis is premature, as there are no data presented to show that the oocytes in CS180-fed animals are as functional as those from OP50-fed animals. Is there a reduction in cross-progeny from CS180-fed hermaphrodites compared to from those fed OP50?

2. Lines 173-181 (Figures 4B and 4D): Two issues here. First, the daf-16 data presented in panels B and D differ. Which is correct? Second, daf-16;daf-2 double mutants should be tested; if daf-16 does not suppress the daf-2 phenotype, it might indicate that this is a daf-16-independent function of daf-2, which would be of interest.

3. Lines 186-188 (Figure 4D): the daf-16;ins-6 data suggest that the roles of ins-6 and daf-16 in this process are complex. daf-16 is epistatic to ins-6, but daf-16 inactivation does not restore the food type difference in progeny that is observed in wild-type animals, as one might expect if INS-6 were simply inhibiting DAF-16 by activating DAF-2. This should be discussed.

4. Lines 208-209 (Figure 5A): is the ASJ ablation phenotype daf-16-dependent?

5. Lines 224-226 (Figure 5C): is the ins-6 mutant phenotype daf-16-dependent?

*Reviewer #2 (Recommendations for the authors):*

1. CS180/OP50 mixing experiments could help indicate whether the onset of oogenesis phenotype, for example, is caused by the presence or absence of something in CS180. But even then, relative to a number of studies that have better defined what microbial determinants induce host changes, the current study is lacking in molecular insight.

2. The daf-16 loss-of-function phenotype and suppression of daf-2 and ins-6 mutant phenotypes could be performed for the lin-41::GFP expression phenotype.

3. If the authors insist on arguing that ins-6 expression in ASJ is induced by CS180 vs. OP50 to cause the observed phenotype, which as noted above seems somewhat implausible, then a strain carrying an ASJ-specific loss of ins-6 expression could be generated to provide support for the authors' model, both with respect to the involvement of ins-6 and ASJ in the observed phenotype.

*Reviewer #3 (Recommendations for the authors):*

I'd like the authors to address two issues:

1) The authors test only a very limited set of bacteria and do the experiments at a single temperature (25C), making the generality of these effects hard to extrapolate. In my opinion, it would be interesting to see whether the food type dependent effects on brood size are mirrored in a consistent manner by other K and B type *E. coli* strains from the *C. elegans* microbiome collection, or by microbiome strains predicted to correspond to LPS types equivalent to those studied here. ASJ is a known polymodal neuron, activated not only by water soluble cues possibly derived from bacterial LPS but also by temperature (with activity increased at higher temperatures). So, it would be interesting to determine if the observed food-type dependent effects in wild type worms are temperature dependent or independent (for example by testing at 20C and 25C).

2) There is some variability in the assays measuring the fraction of worms expressing lin-41::GFP. Part of the issue is that the "absolute time" the germ cells spent developing into sperm before switching to oocytes was measured indirectly in the assay by measuring at "developmental milestones" in the soma (ie reaching a phenotypic milestone in the development the vulva) and not absolute time (eg hours post hatching, or hours post molt). I would have preferred the analysis to treat the data as a cumulative distribution of fraction expressing vs time and the comparison to focus on whether there is a shift in the distributions to later time points. But redoing the experiments as longitudinal measurements would be very laborious and unlikely to yield much of a difference. In any case, that shift should result in a change in the proportion of germ cells that developed into sperm, as shown for wild type in figure 1E. The number of sperm is the biologically relevant endpoint measurement, while the fraction expressing ling-41::GFP at some timepoints could be influenced by other variables, such as length of development (which is affected also by insulin signaling: extended in daf-2 mutants). The "total sperm number before fertilization begins" measurements were not done in the ins-6, odr-7 and would be very informative in testing the model that was proposed. In addition, it would be nice to get sperm counts in the daf-16 and daf-2 mutants, which did not fit quite neatly as ins-6 and odr-7 into the proposed division of labor between ASJ and AWA.

---

## [Author Response]

Essential revisions:1. Provide more insight into bacterial factors that might be contributing to the CS180 oogenesis phenotype;

We thank the Reviewers for this suggestion. Because CS180 is an *E. coli* K-12 strain, we also tested the effects of another *E. coli* K-12 strain on *C. elegans* oogenesis onset. Interestingly, the *E. coli* K-12 HT115 does not elicit the same response as CS180. Unlike the CS180 diet, but like the *E. coli* OP50 diet, HT115 does not promote early oogenesis in wild-type worms. This is shown in Figure 2E of the revised manuscript. Together our data suggest that the bacterial derived cue(s) that promote early oogenesis are specific to CS180 and its derivative CS2429 (Figure 2C).

To start elucidating the nature of the CS180 cue(s), we filtered CS180-conditioned LB media through a nylon membrane that has a pore size of 0.45 µm. The resultant filtrate was added to worms fed OP50. However, supplementation of OP50 with the CS180 filtrate had no effect on worm oogenesis onset (Figure 2G), which shows that the CS180-derived oogenic-inducing cue(s) were lost upon filtration. This also means that the CS180 cues are unlikely to be free, small metabolites. While the identification of these cues are important, it is beyond the scope of the current manuscript.

2. Perform oogenesis experiments using an additional readout distinct from the lin-41::GFP reporter;

We also thank the Reviewers for this suggestion. Thus, we used a second oogenesis marker, *oma-1::GFP* (Lin, Dev Biol 2003, vol 258, pp 226-239). The *oma-1* gene is expressed later than *lin-41* in the differentiating oocytes (Detwiler et al., Dev Cell 2001, vol 1, pp 187-199; reviewed by Albarqi and Ryder, Front Cell Dev Biol 2023, vol 10, 1094295). Through *oma-1::GFP*, we confirmed that wild-type worms fed CS180 underwent earlier oogenesis at mid L4, in contrast to worms fed OP50 (boxed panel in Figure 1I). This observation is consistent with what we saw when wild-type worms express *lin-41::GFP* (boxed panel in Figure 1H).

Moreover, when we ablated the ASJ neurons or deleted the *ins-6* gene in worms expressing *oma-1::GFP*, we saw delayed oogenesis on CS180 (please compare boxed panels in Figure 5E), similar to what we previously observed in *lin-41::GFP*-expressing worms lacking their ASJ neurons or the *ins-6* locus (please compare boxed panels in Figure 5A, D, and F). Together our data support the model that ASJ and *ins-6* are important in promoting early oogenesis at mid L4 when animals are fed an *E. coli* CS180 diet.

3. Strengthen evidence in support the roles of ASJ and INS-6 in mediating the CS180 early oogenesis phenotype (e.g., ASJ-specific deletion of ins-6);

There is strong evidence from several labs, Zhang’s, Buelow’s and ours, that the *ins-6* expressed from ASI has a different function from the *ins-6* expressed in ASJ (Chen et al., Neuron 2013, vol 77, pp 572-585; Tang et al., Curr Biol 2023, vol 33, pp 3835-3850; and this work). Using the same transgenes that we used in our cell-specific rescue experiments for oogenesis onset, the Zhang lab (Chen et al., 2013) showed that INS-6 from ASI, but not ASJ, rescues the aversive learning phenotype of *ins-6* mutants. We showed a different cell-specific requirement for INS-6 in oogenesis: INS-6 from ASJ, but not ASI, rescues the CS180-dependent oogenesis onset phenotype of *ins-6* mutants (Figure 6B). Recently, findings from the Buelow lab further supported the functional differences between the INS-6 peptides produced from ASI and ASJ (Tang et al., 2023). Tang et al. (2023) showed that *ins-6* from ASJ, and not from ASI, alters the neural circuit that promotes salt-dependent associative learning in adult worms. This could be a general mechanism, since there are published examples in the literature where a secreted peptide shows rescue when expressed from one cell or tissue but not from another (e.g., for the insulin-like peptide INS-1; see Kodama et al., Genes Dev 2006, vol 20, pp 2855-2960; Tomioka et al., Neuron 2006, vol 51, pp 613-625). While the mechanism(s) underlying the functional differences between the INS-6 peptides produced by ASI versus ASJ remain unclear, the neuron-specific activities of INS-6 are undeniable and supported by our data and others in the literature.

In addition to showing in this manuscript that it is the ASJ-specific expression of *ins-6* that rescues the oogenesis phenotype of *ins-6* mutants on CS180 (Figure 6B), we now removed *ins-6* specifically from ASJ, but not from ASI. During the decision period for oogenesis onset, around the mid-to-late L3 larval stage (Figure 3B), we previously showed that *ins-6* is only expressed in ASI and ASJ neurons, but only ASJ show increased *ins-6* levels on CS180 (Figure 6A). In the revised manuscript, we showed that ASJ-specific deletion of *ins-6* led to a delay in oogenesis onset on the CS180 diet (Figure 6C). Thus, we demonstrated that ASJ-expressed *ins-6* is both sufficient and necessary to promote early oogenesis on CS180.

4. Address the issue of aberrant kinetics of lin-41::GFP expression in the ASJ ablation strain (potentially with a different oogenesis readout).

ASJ expresses multiple signals (Taylor et al., Cell 2021, vol 184, pp 4329-4347), some of which might also regulate the multiple functions of *lin-41::GFP* in oogenesis. With this in mind, it is to be expected that loss of ASJ might have a more dramatic effect on oogenesis than the deletion of *ins-6* alone, which is a more precise experimental manipulation than eliminating all ASJ-expressed signals.

Following the reviewers’ recommendation, we did use a second oogenesis marker, *oma-1::GFP* (Lin, 2003; please see our response above, Essential Revisions, comment 2). Based on the temporal expression of the two oogenesis markers, *lin-41::GFP* and *oma-1::GFP*, our data support the model that ASJ and *ins-6* are important in promoting early oogenesis at mid L4 when animals are fed *E. coli* CS180.

The stronger phenotype observed with *lin-41::GFP* expression than with *oma-1::GFP* expression upon ablation of ASJ (please compare Figure 5A and E) might be explained by the different temporal requirements and functions of these two genes (Detwiler et al., 2001; Albarqi and Ryder, 2023). Not only is *lin-41* expressed prior to *oma-1* during oogenesis, but *oma-1* also acts redundantly with another RNA-binding protein *oma-2* (Albarqi and Ryder, 2023). Thus, it is possible that multiple signals in ASJ regulate the multiple functions of *lin-41* during oocyte development.

Reviewer #1 (Recommendations for the authors):1. Lines 84-85: I think the conclusion that the decreased progeny phenotype on CS180 is a readout for oogenesis is premature, as there are no data presented to show that the oocytes in CS180-fed animals are as functional as those from OP50-fed animals. Is there a reduction in cross-progeny from CS180-fed hermaphrodites compared to from those fed OP50?

We agree with the Reviewer that decreased progeny on CS180 might be a readout for more than oogenesis onset. Thus, we qualified this statement in the manuscript by presenting it as a hypothesis that total progeny can reflect earlier oogenesis on CS180 (lines 89-90). Indeed, we later showed that the two phenotypes can be uncoupled, since the mechanism(s) that regulate total progeny are not the same as those that regulate oogenesis onset. First, both *E. coli* K-12 bacteria, CS180 and HT115, reduce the total progeny of wild-type worms, in contrast to the *E. coli* B strain OP50 (Figure 2—supplement 1). However, of the three bacterial strains, only CS180 elicited earlier oogenesis (Figure 2E). Second, the loss of food-type differences in the total progeny of *ins-6* mutants was restored when *daf-16* was also deleted (Figure 4B, D, and E). This suggested that the effect of *ins-6* on this phenotype is *daf-16*-dependent. Yet, we showed that to promote early oogenesis, *ins-6* acts independent of *daf-16* (Figure 5F), highlighting the mechanistic differences that underlie the two phenotypes.

Are oocytes of CS180-fed animals more functional or less functional than the oocytes of OP50-fed animals? While we have not assessed the cross-progeny of CS180-fed animals versus those of OP50-fed animals, which is beyond the scope of this manuscript, we did show that the oocytes of CS180-fed animals are fertilized faster than oocytes of OP50-fed animals (Figure 1D). Animals on CS180 produced more live progeny within the first 24 hours of adulthood than animals on OP50 (please see “Measuring the rates of fertilization” in the Materials and methods section). This would argue against a reduction in functionality of the oocytes of CS180-fed animals.

2. Lines 173-181 (Figures 4B and 4D): Two issues here. First, the daf-16 data presented in panels B and D differ. Which is correct? Second, daf-16;daf-2 double mutants should be tested; if daf-16 does not suppress the daf-2 phenotype, it might indicate that this is a daf-16-independent function of daf-2, which would be of interest.

We appreciate the Reviewer’s comments. After multiple trials, we found that the number of progeny of *daf-16* mutants is variable (Figure 4B and E, Figure 4—supplement 1). However, *daf-16* mutants do exhibit food-type differences in their number of progeny (Figure 4B and E). In this manuscript, we focused on the epistasis analysis between *ins-6* and *daf-16* (please see below). The DAF-2 receptor can bind multiple ILP ligands from multiple cells, some of which might be *daf-16*-dependent, while others might be *daf-16*-independent, in influencing an animal’s number of progeny. For an explanation on the variability of the *daf-16* phenotype, please see our response to comment 3 below.

3. Lines 186-188 (Figure 4D): the daf-16;ins-6 data suggest that the roles of ins-6 and daf-16 in this process are complex. daf-16 is epistatic to ins-6, but daf-16 inactivation does not restore the food type difference in progeny that is observed in wild-type animals, as one might expect if INS-6 were simply inhibiting DAF-16 by activating DAF-2. This should be discussed.

As mentioned in our response to comment 1 of the Reviewer, the food type-dependent effect of *ins-6* on total progeny requires *daf-16*. The loss of food-type differences in the number of progeny of *ins-6* mutants is restored when *daf-16* was also removed in these animals (Figure 4E). On the other hand, we observed that *ins-6* acts independent of *daf-16* in promoting early oogenesis on CS180 (Figure 5F). A worm’s number of progeny will be affected by the timing of the spermatogenesis-to-oogenesis switch and by successful fertilization. Thus, *daf-16-*dependent and *daf-16*-independent mechanisms that regulate both aspects of oocyte biology and that can respond to different environmental cues might explain the variability in the total progeny phenotype of *daf-16* mutants. We discuss this in “Oocyte development requires distinct insulin-like signaling mechanisms” in the Discussion section.

4. Lines 208-209 (Figure 5A): is the ASJ ablation phenotype daf-16-dependent?

This is an interesting question. However, because ASJ does express multiple signals besides *ins-6* (Taylor et al., 2021), several which might also affect oogenesis, we focused on testing the *daf-16*-dependence of *ins-6* mutants during oogenesis onset.

5. Lines 224-226 (Figure 5C): is the ins-6 mutant phenotype daf-16-dependent?

No, the *ins-6* mutant oogenesis phenotype is independent of *daf-16*. Please see our responses above.

Reviewer #2 (Recommendations for the authors):1. CS180/OP50 mixing experiments could help indicate whether the onset of oogenesis phenotype, for example, is caused by the presence or absence of something in CS180. But even then, relative to a number of studies that have better defined what microbial determinants induce host changes, the current study is lacking in molecular insight.

Our study focused on how specific sensory neurons mediate the effects of different bacterial diets on three different aspects of *C. elegans* reproductive physiology—total progeny, oogenesis onset and fertilization rates. We examined the effects of three different bacteria, *E. coli* OP50, CS180 and CS2429, on these three phenotypes and the effects of two *Serratia marcescens* strains, Db11 and Db1140, on oogenesis onset. Of these five bacteria, only CS180 and its derivative CS2429, promote early *C. elegans* oogenesis.

In the revised manuscript, we included the effects of a fourth *E. coli* strain, the K-12 HT115 on total progeny (Figure 2—supplement 1), oogenesis onset (Figure 2E) and fertilization rates (Figure 2F). We found that HT115 does not elicit the same response as CS180 on oogenesis onset and fertilization rates. Thus, the oogenic-inducing and fertilization-enhancing cue(s) appear to be specific to CS180 and its derivative CS2429. We started characterizing the potential nature of these CS180-derived cue(s). So far, we found that these cues are unlikely to be free, small metabolites, since they were lost upon filtration of the CS180-conditioned LB media through a nylon membrane that has a pore size of 0.45 µm (Figure 2G and H). While we agree with the Reviewer that the identification of these cues are important, we believe that it is beyond the scope of this manuscript.

More importantly, we showed that the sensory neuron ASJ does modulate the timing of oogenesis and that this involves the insulin-like peptide *ins-6* (please see our responses to the Essential Revisions section and Figures 5 and 6). We also showed that ASJ (Figure 7G and K) or *ins-6* (Figure 8D) does not affect the food type-dependent fertilization rates, which are modulated by a different sensory neuron, the olfactory neuron AWA (Figure 7J and K). AWA in turn has no effect on the timing of oogenesis (Figure 7L). Thus, this manuscript links specific sensory neurons and insulin-like peptides to distinct aspects of oocyte biology, which we believe is a significant advance in the field of reproductive biology.

2. The daf-16 loss-of-function phenotype and suppression of daf-2 and ins-6 mutant phenotypes could be performed for the lin-41::GFP expression phenotype.

We conducted an epistasis analysis between *ins-6* and *daf-16* with regard to early oogenesis onset on the CS180 diet. Through recombination of *lin-41::GFP* with the *daf-16* deletion mutation on chromosome I, we showed that *daf-16* mutants exhibit early oogenesis at mid L4 on CS180 (Figure 5C and F), which is unlike the *ins-6* deletion (null) mutants or the reduction-of-function mutations in *daf-2*. Both *ins-6* and *daf-2* mutants exhibit delayed oogenesis on CS180 (Figure 5B, D, and F). Interestingly, the delayed oogenesis phenotype of *ins-6* null mutants was not rescued by loss of *daf-16*, suggesting that wild-type *ins-6* promotes early oogenesis independent of *daf-16* (Figure 5F). This is reminiscent of the Arur lab’s findings, where *daf-2* promotes germline meiotic progression independent of *daf-16* in response to food availability (Lopez et al., Dev Cell 2013, vol 27, pp 227-240).

3. If the authors insist on arguing that ins-6 expression in ASJ is induced by CS180 vs. OP50 to cause the observed phenotype, which as noted above seems somewhat implausible, then a strain carrying an ASJ-specific loss of ins-6 expression could be generated to provide support for the authors' model, both with respect to the involvement of ins-6 and ASJ in the observed phenotype.

We address this in Essential revisions, point 3. Briefly, we disagree that this is puzzling, since several labs have already shown that there are functional differences between the INS-6 produced from ASI versus the INS-6 produced from ASJ, using different experimental approaches (Chen et al., 2013; Tang et al., 2023; and this work). Indeed, the cell-specific activities of a secreted signal is not limited to INS-6, but has also been described for other secreted peptides, such as INS-1 (Kodama et al., 2006; Tomioka et al., 2006; Takeishi et al., *eLife* 2020, vol 9, e61167). Thus, the interesting question is why functional differences exist between the INS-6 peptides from the two neurons. This is a fascinating question, but beyond the scope of this manuscript.

We disagree that [the *ins-6* change in expression in ASJ] is modest and that the oogenic effect of such a change is implausible.

First, the change in *ins-6p::mCherry* expression in ASJ on CS180 is comparable to other physiologically-important expression changes that have been reported for other genes (for example, Entchev et al., *eLife* 2015, vol 4, 4:e06259, for the tryptophan hydroxylase *tph-1* and the TGF-β *daf-7*; and Tataridas-Pallas et al., PLoS Genet 2021, vol 17, e1009358, for the neuronally expressed NRF transcription factor *skn-1b*). Second, it is worth noting that we were using a single-copy reporter for *ins-6* expression, where detected changes will be smaller but should be closer to physiological responses. It is possible that multiple-copy reporters will give larger changes, but that would be further from a physiological response. Third, the change in *ins-6p::mCherry* expression is comparable in scale to the *ins-6* mutant phenotype. Our results showed that the 35% increase in ASJ expression of *ins-6* is due to food type (Figure 6A; mean fluorescence on OP50 = 1526 + 94; mean fluorescence on CS180 = 2056 + 104). This change in magnitude is similar to the loss of *lin-41::GFP* expression in mid L4 of *ins-6* mutants versus controls. About 30% to 43% of control worms express *lin-41::GFP*, whereas 0% of *ins-6* mutants express the same reporter at mid L4 on CS180 (Figure 5 and its associated supplement).

Reviewer #3 (Recommendations for the authors):I'd like the authors to address two issues:1) The authors test only a very limited set of bacteria and do the experiments at a single temperature (25C), making the generality of these effects hard to extrapolate. In my opinion, it would be interesting to see whether the food type dependent effects on brood size are mirrored in a consistent manner by other K and B type *E. coli* strains from the *C. elegans* microbiome collection, or by microbiome strains predicted to correspond to LPS types equivalent to those studied here. ASJ is a known polymodal neuron, activated not only by water soluble cues possibly derived from bacterial LPS but also by temperature (with activity increased at higher temperatures). So, it would be interesting to determine if the observed food-type dependent effects in wild type worms are temperature dependent or independent (for example by testing at 20C and 25C).

We partly address this in Essential revisions, point 1.

We also appreciate the Reviewer’s comments about the temperature-dependence of our phenotypes. We are beginning to explore this aspect, which we believe is beyond the scope of this manuscript.

2) There is some variability in the assays measuring the fraction of worms expressing lin-41::GFP. Part of the issue is that the "absolute time" the germ cells spent developing into sperm before switching to oocytes was measured indirectly in the assay by measuring at "developmental milestones" in the soma (ie reaching a phenotypic milestone in the development the vulva) and not absolute time (eg hours post hatching, or hours post molt). I would have preferred the analysis to treat the data as a cumulative distribution of fraction expressing vs time and the comparison to focus on whether there is a shift in the distributions to later time points. But redoing the experiments as longitudinal measurements would be very laborious and unlikely to yield much of a difference.

We appreciate the Reviewer’s comments. At 25^o^C, the mid L4 substage is 41-43 h; late L4, 44-46 h; and lethargus, 46-48 h after egg laying. However, we think that normalization of the oogenesis reporter expression against the vulval structure-based substages that were defined by Mok et al. (BMC Dev Biol 2015, vol 15, 26) is more precise.

In any case, that shift should result in a change in the proportion of germ cells that developed into sperm, as shown for wild type in figure 1E. The number of sperm is the biologically relevant endpoint measurement, while the fraction expressing ling-41::GFP at some timepoints could be influenced by other variables, such as length of development (which is affected also by insulin signaling: extended in daf-2 mutants). The "total sperm number before fertilization begins" measurements were not done in the ins-6, odr-7 and would be very informative in testing the model that was proposed. In addition, it would be nice to get sperm counts in the daf-16 and daf-2 mutants, which did not fit quite neatly as ins-6 and odr-7 into the proposed division of labor between ASJ and AWA.

By linking specific sensory neurons and insulin-like peptides to distinct aspects of oocyte biology, we have laid the groundwork for future studies. These include exploring the effects of the bacterial diet on *C. elegans* sperm, which we believe is beyond the scope of the current manuscript.